



**Spatial-temporal patterns of inorganic nitrogen air concentrations**
**and deposition in eastern China**
Wen Xu[1,2], Lei Liu[3], Miaomiao Cheng[4], Yuanhong Zhao[5], Lin Zhang[5], Yuepeng Pan[6],
Xiuming Zhang[7], Baojing Gu[7], Yi Li[8], Xiuying Zhang[3], Jianlin Shen[9], Li Lu[10],
Xiaosheng Luo[11], Yu Zhao[12], Zhaozhong Feng[2*], Jeffrey L. Collett, Jr.[13], Fusuo
Zhang[1], Xuejun Liu[1*]
[1]College of Resources and Environmental Sciences, Key Laboratory of Plant-Soil Interactions of
MOE, Beijing Key Laboratory of Cropland Pollution Control and Remediation, China
Agricultural University, Beijing 100193, China
[2]State Key Laboratory of Urban and Regional Ecology, Research Center for Eco-Environmental
Sciences, Chinese Academy of Sciences, Shuangqing Road 18, Haidian District, Beijing, 100085,
China
[3]Jiangsu Provincial Key Laboratory of Geographic Information Science and Technology,
International Institute for Earth System Science, Nanjing University, Nanjing, 210023, China
[4]State Key Laboratory of Environmental Criteria and Risk Assessment, Chinese Research
Academy of Environmental Sciences, Beijing 100012, China
[5]Laboratory for Climate and Ocean-Atmosphere Sciences, Department of Atmospheric and
Oceanic Sciences, School of Physics, Peking University, Beijing 100871, China
[6]State Key Laboratory of Atmospheric Boundary Layer Physics and Atmospheric Chemistry
(LAPC), Institute of Atmospheric Physics, Chinese Academy of Sciences, Beijing, 100029, China
[7]Department of Land Management, Zhejiang University, Hangzhou 310058, People's Republic of
China
[8]Arizona Department of Environmental Quality, Phoenix, AZ 85007, USA
[9]Institute of Subtropical Agriculture, Chinese Academy of Sciences, Changsha 4410125, China
[10]Institute of Surface-Earth System Science, Tianjin University, Tianjin, 300072, China
[11]Institute of Plant Nutrition, Resources and Environmental Sciences, Henan Academy of
Agricultural Sciences, Henan Key Laboratory of Agricultural Eco-environment, Zhengzhou,
450002, China
[12]State Key Laboratory of Pollution Control & Resource Reuse, School of the Environment,
Nanjing University, 163 Xianlin Ave., Nanjing, Jiangsu 210023, China
[13]Department of Atmospheric Science, Colorado State University, Fort Collins, Colorado, 80523
USA
[*]Correspondence to: X. J. Liu (liu310@cau.edu.cn) and Z.Z. Feng (fzz@rcees.ac.cn)



**Abstract:**

Five-year (2011-2015) measurements of gaseous $NH_3$, $NO_2$ and $HNO_3$ and particulate
$NH_4^+$ and $NO_3^-$ in air and/or precipitation were conducted at twenty-seven sites in a
Nationwide Nitrogen Deposition Monitoring Network (NNDMN) to better understand
spatial and temporal (seasonal and annual) characteristics of reactive nitrogen ($N_r$)
concentrations and deposition in eastern China. Our observations reveal annual
average concentrations (16.4-32.6 µg N m$^{-3}$), dry deposition fluxes (15.8-31.7 kg N
ha$^{-1}$ yr$^{-1}$) and wet/bulk deposition fluxes (18.4-28.0 kg N ha$^{-1}$ yr$^{-1}$) based on land use
were ranked as urban > rural > background sites. Annual concentrations and dry
deposition fluxes of each $N_r$ species in air were comparable at urban and background
sites in northern and southern regions, but were significantly higher at northern rural
sites. These results, together with good agreement between spatial distributions of
$NH_3$ and $NO_2$ concentrations determined from ground measurements and satellite
observations, demonstrate that atmospheric $N_r$ pollution is heavier in the northern
region than in the southern region. No significant inter-annual trends were found in
the annual $N_r$ dry and wet/bulk N deposition at almost all of the selected sites. A lack
of significant changes in annual averages between the 2013-2015 and 2011-2012
periods for all land use types, suggests that any effects of current emission controls
are not yet apparent in $N_r$ pollution and deposition in the region. Ambient
concentrations of total $N_r$ exhibited a non-significant seasonal variation at all land use
types, although significant seasonal variations were found for individual $N_r$ species
(e.g., $NH_3$, $NO_2$ and $pNO_3^-$) in most cases. In contrast, dry deposition of total $N_r$
exhibited a consistent and significant seasonal variation at all land use types, with the
highest fluxes in summer and the lowest in winter. Based on sensitivity tests by the
GEOS-Chem model, we found that $NH_3$ emissions from fertilizer use (including
chemical and organic fertilizers) were the largest contributor (36%) to total inorganic
$N_r$ deposition over eastern China. Our results not only improve the understanding of
spatial-temporal variations of $N_r$ concentrations and deposition in this pollution
hotspot, but also provide useful information for policy-makers that mitigation of $NH_3$
emissions should be a priority to tackle serious N deposition in eastern China.



## 1. Introduction

In China, and globally, human activities have dramatically increased emissions of nitrogen oxides ($NO_x$=NO+$NO_2$) and ammonia ($NH_3$) into the atmosphere since the beginning of the industrial revolution (Galloway et al., 2008; Liu et al., 2013). $NO_x$ and $NH_3$ emitted to the atmosphere are transformed to nitrogen-containing particles (e.g., particulate $NH_4^+$ and $NO_3^-$, and organic nitrogen) (Ianniello et al., 2010; Zhang et al., 2015), which are major chemical constituents of airborne $PM_{2.5}$ (particulate matter with a diameter of 2.5 μm or less) and have implications for air quality and climate (Fuzzi et al., 2015). As a result of elevated $N_r$ emissions, nitrogen (N) deposition through dry and wet processes has also substantially increased over China (Liu et al., 2013; Lu et al., 2007, 2014; Jia et al., 2014, 2016), and excessive deposition of N has resulted in detrimental impacts including decreased biological diversity (Bobbink et al., 2010), nutrient imbalance (Li et al., 2016), increased soil acidification (Yang et al., 2015), eutrophication of water bodies (Fenn et al., 2003), and increased greenhouse gas emissions (Gruber and Galloway, 2008). Furthermore, $N_r$-associated haze pollution episodes, characterized by high concentrations of $PM_{2.5}$, occur frequently in China, as evidenced in particular in 2013 (Guo et al., 2014; Huang et al., 2014; Tian et al., 2014).

In order to control its notorious air pollution, China has reduced national emissions of $SO_2$ and particulate matter by 14% and 30%, respectively, from 2005 to 2010 (MEPC, 2011). Additionally, stringent measures (e.g., using selective catalytic/non-catalytic reduction systems, and implementing tighter vehicle emission standards) were implemented during the 12th Five Year Plan (FYP) period (2011-2015), with aims to reduce 2015 annual emissions of $SO_2$ and $NO_x$ by 8% and 10%, respectively, relative to 2010 levels (Xia et al., 2016). However, there is as yet no regulation or legislation that deals with national $NH_3$ emissions and thus emission reductions of $SO_2$ and $NO_x$ to achieve desired air-quality improvement goals will be compromised (Gu et al., 2014). Significant increases in $PM_{2.5}$ concentrations have been observed in the years 2013 and 2014 as compared to 2012, excluding the influence of meteorological conditions on inter-annual variations (Liang et al., 2015).



Other studies with more conclusive evidence have likewise suggested that $NH_3$ plays
a vital role in sulfate formation and exacerbates severe haze pollution development in
urban regions of China (Wang et al., 2016), even acting as the key limiting factor for
the formation of secondary inorganic aerosol (Wu et al., 2016). In addition, due to
higher local and regional concentrations of $NH_3$ in the atmosphere, nitrate-driven haze
pollution occurred during summertime in urban environment in the North China Plain
(Li et al., 2018). The absolute and relative concentrations of particulate nitrate in
urban Beijing increased with haze development (Pan et al., 2016). Also, nitrate
contributed to a large fraction of the elevated $PM_{2.5}$ concentrations at a rural site in the
North China Plain and high $NH_3$ in the early morning accelerated the formation of
fine nitrates (Wen et al., 2015).
High rates of N deposition have also been observed during 2011-2014 across
China (Xu et al., 2015). However, to date no study, based on long-term ground-based
observations, has provided any information on the effectiveness of $SO_2$ and $NO_x$
emission controls on N deposition in China. Non-linearities have been identified
between reductions in emission and deposition in Europe over the last 3 decades
(Aguillaume et al., 2016; Fowler et al., 2007). Due to the tightly coupled yet complex
relationship between emissions, concentrations and deposition, long-term monitoring
networks can provide a test of the effectiveness of emission controls (Erisman et al.,
2003). Currently two national N deposition networks are operational in China, i.e. the
Nationwide Nitrogen Deposition Monitoring Network (NNDMN, Liu et al., 2011; Xu
et al., 2015) and the Chinese Ecosystem Research Network (CERS, Zhu et al., 2015).
The NNDMN containing 43 *in situ* monitoring sites has been operational since 2010
to measure wet N deposition and ambient concentrations of five major $N_r$ species (i.e.,
gaseous $NH_3$, $NO_2$ and $HNO_3$, and particulate $NH_4^+$ and $NO_3^-$), the latter for
subsequence estimation of dry deposition. The CERS was established in 1988 and
mainly focused on wet N deposition at 41 field stations. In addition to ground-based
measurements, satellite observations enable retrieval of atmospheric $NH_3$ and $NO_2$
with high temporal and spatial resolutions (Dammer et al., 2016; Russell et al., 2012),
providing a means to reveal spatial distributions and long-term trends of ambient $NH_3$





and $NO_2$ levels at regional to global scales, and also to evaluate the effectiveness of
emission controls (Krotkov et al., 2016). However, to effectively use the vast satellite
data sets for environmental monitoring, it is critical to validate these remote sensing
observations using *in situ* surface observations (Pinder et al., 2011; Van Damme et al.,

2015).

Eastern China is a developed region with the largest densities of population,

economic activity and resource consumption in the country (He et al., 2015). Recent
satellite observations indicate that tropospheric $NH_3$ and $NO_2$ levels in eastern China
were both much greater than other regions of the world from 2005-2015 (Demmer et
al., 2016; Krotkov et al., 2016). Accordingly, this region received the highest levels of
dry N deposition in the world (Vet et al., 2014), and was regarded as a primary export
region of N deposition for neighboring countries (Ge et al., 2014). Based on
meta-analysis of published observations, some studies have provided information on
the magnitudes, spatial distributions, and decadal variations of wet/bulk N deposition
in China (Liu et al., 2013; Jia et al., 2014), but the analyzed data were limited to time
periods between 1980 and 2010. Although a recent study (Jia et al., 2016) has
reported a clear increasing trend of dry N deposition in eastern China between 2005
and 2014, considerable uncertainty may exist due to estimates of gaseous $HNO_3$ and
particulate $NH_4^+$ and $NO_3^-$ ($pNH_4^+$ and $pNO_3^-$) concentrations using $NO_2$ satellite data,
which is in part manifested by Liu et al. (2017a). Furthermore, seasonal patterns of $N_r$
concentrations and deposition have not yet been systematically investigated at a large
spatial scale in this region, although spatial patterns of dry $N_r$ deposition for
representative months of four seasons (i.e., January for winter, April for spring, July
for summer, October for autumn) in 2010 have been mapped with the RAMS-CMAQ
model (Han et al., 2017). Thus, the spatial and temporal (annual and seasonal)
variations of $N_r$ concentrations, and dry and wet deposition in eastern China require
further exploration using ground-based measurements, especially for time periods
after 2010.

The present study aims to examine spatial-temporal (annual and seasonal)

characteristics of $N_r$ concentrations in air ($NH_3$, $NO_2$, $HNO_3$, $pNH_4^+$ and $pNO_3^-$) and





159 precipitation ($NH_4^+$-N and $NO_3^-$-N) and their corresponding dry and wet/bulk N

160 deposition, through a 5-year (2011-2015) monitoring period at 27 NNDMN *in situ*

161 sites in eastern China. In addition, we compare spatial-temporal variability of

162 measured $NH_3$ and $NO_2$ concentrations with variations of the corresponding satellite

163 retrieval columns, as well as inter-annual trends in $N_r$ deposition and emissions.

164 Finally, emission sources contributing to total N deposition over eastern China are

165 examined.

166 **2. Materials and methods**

167 **2.1 Study area and site descriptions**

168   The present study was conducted in eastern China, which is distinguished by the

169 "Hu Line" (She, 1998). This region has spatial heterogeneity in levels of economic

170 development, resulting in significant spatial differences in $NH_3$ and $NO_x$ emissions

171 (Fig. 1b and c). Thus, to better analyze spatial and temporal variabilities in measured

172 $N_r$ concentrations and deposition, we divided eastern China into northern and southern

173 regions using the Qinling Mountains-Huaihe River line (Fig. 1a), of which the

174 division basin was based on the differences in natural conditions, agricultural

175 production, geographical features and living customs. As for specific differentiations,

176 for example, the northern region adopted a centralized domestic heating policy for

177 late autumn and winter seasons but the south has not; annual average precipitation

178 amounts were generally greater than 800 mm in the south but were less than 800 mm

179 in the north. In addition, the north is characterized by calcareous soil, which could

180 result in higher soil $NH_3$ volatilization (Huang et al., 2015), vs. the acidic red soil in

181 the south.

182   The NNDMN was operated in line with international standards by China

183 Agricultural University (CAU); 35 NNDMN sites were located in eastern China (Xu

184 et al., 2015). For our analysis, we considered twenty-seven sites in total, with most

185 having continuous data covering 5 years: 13 sites were located north of the Qinling

186 Mountains-Huaihe River line (China Agricultural University-CAU, Zhengzhou-ZZ,

187 Dalian-DL, Shuangzhuang-SZ, Quzhou-QZ, Yangqu-YQ, Zhumadian-ZMD,

188 Yanglin-YL, Yucheng-YC, Gongzhulin-GZL, Lishu-LS, Lingshandao-LSD,



Changdao-CD), and 14 sites were located south of the line (Nanjing-NJ, Baiyun-BY,
Wenjiang-WJ, Wuxue-WX, Taojing-TJ, Fengyang-FY, Zhanjiang-ZJ, Fuzhou-FZ,
Fenghua-FH, Ziyang-ZY, Yangting-YT, Jiangjin-JJ, Huinong-HN, Xishan-XS).

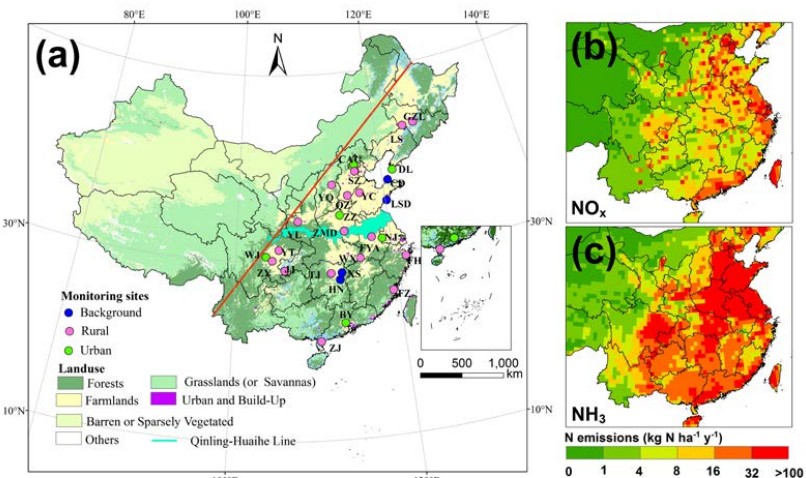

**Figure 1**. Spatial distributions of the 27 monitoring sites (a), $NO_x$ emissions (b)
and $NH_3$ emissions (c) in Eastern China ($NH_3$ and $NO_x$ emission data were for the
year 2010 and obtained from Liu et al. (2017b)).
All the sites are located as far away as possible and practical from local direct
emission sources to increase regional representativeness. They can be divided into
three categories according to their geopolitical location and their proximity to the
main emission sources: urban sites (abbreviated as U), rural sites (cropland areas, R),
and background sites (coastal and forest areas, B). Information on the monitoring sites,
such as land use types, coordinates, and measurement periods are listed in Table S1 of
the Supplement. Detailed descriptions of all the sites including the surrounding
environment and nearby emission sources can be found in Xu et al. (2015).
**2.2 Field sampling and chemical analysis**
Continuous measurements were performed during the period from January 2011
to December 2015 at the 27 study sites, except for eleven sites (ZZ, ZMD, YC, LSD,
NJ, WX, FYA, ZJ, YT, JJ, and HN), where field sampling was carried out after the
year 2010 and/or interrupted during the period due to instrument failure (details in





Table S1, Supplement). Ambient $N_r$ concentrations of gaseous $NH_3$ and $HNO_3$, and
$pNH_4^+$ and $pNO_3^-$ (for which the empirically determined effective size cut-off for
aerosol sampling is of the order of 4.5 µm) were measured using an active DELTA
(DEnuder for Long-Term Atmospheric sampling; Tang et al., 2009) system; gaseous
$NO_2$ was sampled in three replicates with passive diffusion tubes (Gradko
International Limited, UK). The air intakes of the DELTA system and the $NO_2$ tubes
were mounted 2 m above the ground at most sites and protected from precipitation
and direct sunlight with a rigid plastic box and a PVC shelter, respectively. All
measurements of $N_r$ concentration were based on monthly sampling (one sample per
month for each $N_r$ species). Detailed information on measuring methods and
collection are given in Sect. S1 of the Supplement.
To collect precipitation (here termed as wet/bulk deposition, which contains wet
and some dry deposition due to the use of an open sampler) samples, a standard
precipitation gauge (SDM6, Tianjin Weather Equipment Inc., China) was
continuously exposed beside the DELTA system (ca. 2 m). Immediately after each
precipitation event (08:00–08:00 next day, Greenwich Mean Time +8), samples
(including rain and melted snow) were collected and stored in clean polyethylene
bottles (50 mL) at -18 $^{o}C$ until sent to the CAU laboratory for analysis. Each collector
was rinsed three times with high-purity water after each collection and once every
week to limit contamination from accumulated dust.
In the analytical laboratory, acid-coated denuders and aerosol filters were
extracted with 6 and 10 mL of high-purity water (18.2 MΩ), respectively, and
analyzed for $NH_4^+$-N with an AA3 continuous-flow analyzer (CFA) (BranC Luebbe
GmbH, Norderstedt, Germany). Carbonate-coated denuders and filters were both
extracted with 10 mL 0.05% $H_2O_2$ solution followed by analysis of $NO_3$-N using the
same CFA. $NO_2$ samples, extracted with a solution containing sulfanilamide, $H_3PO_4$,
and N-1-naphthylethylene-diamine, were determined using a colorimetric method by
absorption at a wavelength of 542 nm (Xu et al., 2016). Precipitation samples were
filtered through a syringe filter (0.45 mm, Tengda Inc., Tianjin, China) and analyzed
for $NH_4^+$-N and $NO_3^-$-N using the CFA as mentioned above. Quality assurance and





quality control procedures adopted in the analytical laboratory are described by Xu et
al. (2017). Further details of precipitation measurement, samples handling, and
chemical analysis are reported in Xu et al. (2015).

**2.3 Deposition estimate**

Wet/bulk deposition of $NH_4^+$-N and $NO_3^-$-N were calculated per month and year
by multiplying the precipitation amount by their respective volume-weighted mean
(VWM) concentrations. The dry deposition flux of gaseous and particulate $N_r$ species
was calculated as the product of measured concentrations by modeled deposition
velocities ($V_d$). The dry deposition velocities of five $N_r$ species were calculated by the
GEOS (Goddard Earth Observing System)-Chem chemical transport model (CTM)
(Bey et al., 2001; http://geos-chem.org), and have been reported in a companion paper
(Xu et al., 2015). In brief, the model calculation of dry deposition of $N_r$ species
follows a standard big-leaf resistance-in-series model as described by Wesely (1989)
for gases and Zhang et al. (2001) for aerosol. We used archived hourly $V_d$ from
January 2011 to May 2013 and filled the gap for the period (from June 2013 to
December 2015) when GEOS meteorological data are unavailable using the mean
values calculated from all the available simulations. The monthly $V_d$ at each site was
averaged from the hourly dataset.

**2.4 Satellite retrievals of $NH_3$ and $NO_2$**

Comparisons between satellite observations and ground-based measurements
were evaluated at the twenty-seven sites in order to accurately examine the
spatial-temporal pattern of $NH_3$ and $NO_2$ concentrations. For $NH_3$, we used the
products retrieved from the Infrared Atmospheric Sounding Interferometer (IASI)
instrument (aboard the MetOp-A platform), which crosses the equator at a mean local
solar time of 9:30 a.m. and 9:30 p.m. The IASI-$NH_3$ product is based on the
calculation of a spectral hyperspectral range index and subsequent conversion to $NH_3$
total columns via a neural network. The details of the IASI-$NH_3$ retrieval method are
described in Whitburn et al. (2016). We only considered the observations from the
morning overpass as they are generally more sensitive to $NH_3$ because of higher
thermal contrast at this time of day (Van Damme et al., 2015; Dammers et al., 2016).





The daily IASI-NH$_3$ data (provided by the Atmospheric Spectroscopy Group at
Université Libre De Bruxelles, data available at http://iasi.aeris-data.fr/NH$_3$/) from 1
January 2011 to 31 December 2015 was used in the present study. We did not use the
IASI_NH$_3$ after 30 September 2014 for the temporal analysis because an update of the
input meteorological data had caused a substantial increase in the retrieved
atmospheric NH$_3$ columns. Only observations with a cloud coverage lower than 25%,
and relative error lower than 100% or absolute error smaller than $5 \times 10^{15}$ molecules
cm$^{-2}$ were processed. The methodology is provided in detail in Liu et al. (2017b). In
brief, all observations were gridded to a 0.5° latitude × 0.5° longitude grid, and then
we calculated the monthly arithmetic mean by averaging the daily values with
observations points within each grid cell. Similarly, we calculated the annual
arithmetic mean by averaging the daily values with observations points within the grid
cell over the whole year.
For NO$_2$ we used the products from the Ozone Monitoring Instrument (OMI)
resided on NASA's EOS-Aura satellite, which was launched in July 2004 into a
sun-synchronous orbit with a local equator crossing time at approximately 1:45 p.m.
OMI detects the backscattered solar radiation from the Earth's atmosphere within the
UV-vis spectral window between 270-500 nm, to achieve nearly global coverage daily,
with a spatial resolution ranging from 13 km × 24 km at nadir to 24 km × 128 km at
the edge of the swath (Russell et al., 2012). We used tropospheric NO$_2$ retrievals from
the DOMINO (Dutch Finnish Ozone Monitoring Instrument) algorithm version 2. The
retrieval algorithm is described in detail in Boersma et al. (2007). The tropospheric
NO$_2$ columns used in this study are monthly means from 1 January 2011 to 30
December 2015 with a spatial resolution of 0.125° latitude × 0.125° longitude (data
available at http://www.temis.nl/airpollution/no2.html).
**2.5 Statistical analysis**
One-way analysis of variance (ANOVA) and two-independent-samples *t* tests
were applied to detect significant differences in seasonal mean concentrations and
deposition fluxes of measured N$_r$ species as well as their annual mean deposition
fluxes for three land use types (rural, urban and background). As there was large



site-to-site variability in annual $N_r$ concentrations and deposition fluxes at monitoring
sites within the same land use types, averaging data into annual values for land use
types is unlikely to be truly representative of actual trends. Thus, annual trends of the
variables were evaluated at a single site scale rather than by land use type. Trend
analysis was conducted using Theil regression (Theil, 1992) and the Mann-Kendall
test (Gilbert, 1987; Marchetto et al., 2013). We defined an increasing (decreasing)
trend as a positive (negative) slope of the Theil regression, while a statistical
significance level ($p<0.01$) of a trend was evaluated by the non-parametric
Mann-Kendall test ($p$ value). Non-parametric methods usually have the advantage of
being insensitive to outliers, and allow missing data and non-normal distribution of
data (Gilbert, 1987; Salmi et al., 2002), appropriate for the analyzed data set. The
Mann-Kendall method is appropriate for detection of monotonic trends in data series
that have no seasonal variation or autocorrelation. Atmospheric concentrations and
deposition fluxes of $N_r$ species, however, generally have distinct seasonal variability
(Pan et al., 2012) and the Mann-Kendall test is thus applied to annual values.

Satellite observations during 2005-2015 indicate that tropospheric $NO_2$ levels

peaked in 2011 over China (Krotkov et al., 2016; Duncan et al., 2016) and $NO_x$
emissions peaked in 2011/2012 (Miyazaki et al., 2017; van der A et al., 2017; Souri et
al., 2017). To assess the impact of emission control measures on measured $N_r$
concentrations and deposition fluxes at different land use types, we compared
arithmetic mean values averaged from the last 3-year period (2013-2015) with those
averaged from the first 2-year period (2011-2012) for monitoring sites with
continuous 5-year measurements (22 sites for dry, and 17 sites for wet/bulk). Seasonal
concentrations and deposition fluxes of measured $N_r$ species were calculated using the
arithmetic average of matched seasons during the sampling periods; spring refers to
March-May, summer covers June-August, autumn refers to September-November,
and winter covers December-February.







## 3. Results

### 3.1 Spatial variability in concentrations of $N_r$ species in air and precipitation



Summary statistics of monthly mean concentrations of $NH_3$, $NO_2$, $HNO_3$, $pNH_4^+$,
and $pNO_3^-$ at the twenty-seven monitoring sites during 2011-2015 are listed in Table
S2 of the Supplement. Monthly mean concentrations of $NH_3$, $NO_2$, $HNO_3$, $pNH_4^+$, and
$pNO_3^-$ ranged from 0.16 (TJ)-39.57 (WJ), 0.55 (LS)-29.06 (WJ), 0.04 (YQ)-4.93
(CAU), 0.11 (ZY)-57.20 (QZ), and 0.01 (DL)-32.06 (ZZ) $\mu g\ N\ m^{-3}$, respectively. On
the basis of geographical location and classification of each site, the annual mean
concentrations of each $N_r$ species were calculated for three land use types in eastern
China and its northern and southern regions (Table 1).
**Table 1.** Annual average (standard error) concentrations of various $N_r$ compounds in
air and precipitation at different land use types in eastern China and its northern and
southern regions for the 5-year period 2011-2015.

| Region[a] | LUY[b] | Ambient conc. $\mu g\ N\ m^{-3}$ | | | | | | Rainwater conc. $mg\ N\ L^{-1}$ | | |
|---|---|---|---|---|---|---|---|---|---|---|
| | | $NH_3$ | $NO_2$ | $HNO_3$ | $pNH_4^+$ | $pNO_3^-$ | Total $N_r$ | $NH_4^+$ | $NO_3^-$ | TIN |
| EC | Urban | 8.5 | 10.2 | 1.6 | 8.2 | 4.0 | 32.6 | 1.6 | 1.9 | 3.5 |
| | (n=6) | (1.4) | (1.0) | (0.2) | (1.8) | (0.8) | (4.1) | (0.3) | (0.2) | (0.5) |
| | Rural | 7.2 | 6.0 | 1.2 | 6.7 | 2.8 | 23.9 | 1.7 | 1.4 | 3.1 |
| | (n=17) | (0.9) | (0.5) | (0.1) | (1.1) | (0.3) | (2.7) | (0.2) | (0.2) | (0.4) |
| | BKD[c] | 3.9 | 5.2 | 0.9 | 4.5 | 1.9 | 16.4 | 1.4 | 1.2 | 2.6 |
| | (n=4) | (0.6) | (0.3) | (0.1) | (0.4) | (0.3) | (1.4) | (0.3) | (0.4) | (0.6) |
| NREC | Urban | 8.1 | 11.7 | 1.6 | 8.6 | 5.1 | 35.1 | 2.2 | 2.4 | 4.6 |
| | (n=3) | (2.4) | (1.6) | (0.3) | (2.3) | (1.4) | (7.7) | (0.4) | (0.2) | (0.4) |
| | Rural | 9.9 | 7.4 | 1.4 | 9.2 | 3.7 | 31.6 | 2.4 | 2.0 | 4.4 |
| | (n=8) | (1.2)** | (0.7)* | (0.1)* | (1.9)* | (0.5)* | (3.8)** | (0.3)** | (0.2)** | (0.4)** |
| | BKD | 4.7 | 5.7 | 1.0 | 5.1 | 2.4 | 18.8 | 1.8 | 1.5 | 3.3 |
| | (n=2) | (0.6) | (0.3) | (0.1) | (0.2) | (0.3) | (0.1) | (0.2) | (0.3) | (0.1) |
| SREC | Urban | 8.9 | 8.7 | 1.6 | 7.9 | 2.9 | 30.1 | 1.1 | 1.5 | 2.6 |
| | (n=3) | (1.8) | (0.6) | (0.1) | (3.1) | (0.2) | (4.5) | (0.3) | (0.3) | (0.6) |
| | Rural | 4.9 | 4.6 | 1.0 | 4.5 | 1.9 | 17.0 | 1.1 | 0.9 | 2.0 |
| | (n=9) | (0.6) | (0.6) | (0.1) | (0.6) | (0.2) | (1.7) | (0.2) | (0.1) | (0.3) |
| | BKD | 3.1 | 4.7 | 0.8 | 4.0 | 1.4 | 14.0 | 1.0 | 0.6 | 1.6 |
| | (n=2) | (0.7) | (0.4) | (0.1) | (0.2) | (0.2) | (0.6) | (0.0) | (0.0) | (0.0) |

[a] EC: eastern China; NREC: northern region of eastern China; SREC: southern region



343 of eastern China. [b] LSY: land use type; n denotes number of monitoring sites. [c] BKD:

344 Background. [*]Significant at the 0.05 probability level. [**]Significant at the 0.01

345 probability level.

346  In eastern China, annual mean concentrations of $NH_3$, $NO_2$, $HNO_3$, $pNH_4^+$, and

347 $pNO_3^-$ at the urban sites ($1.6 \pm 0.2$ to $10.2 \pm 1.0$ µg N m$^{-3}$) were 18-44% and 78-120%

348 higher than their corresponding concentrations at the rural ($1.2 \pm 1.0$ to $7.2 \pm 0.9$ µg N

349 m$^{-3}$) and background ($0.9 \pm 0.1$ to $5.2 \pm 0.3$ µg N m$^{-3}$) sites, respectively. Analogous

350 patterns also occurred for all measured $N_r$ in each region, except for $NH_3$ and $pNH_4^+$

351 in the northern region, for which the mean concentrations were 18% and 7% lower at

352 the urban sites than at the rural sites, respectively.

353  Comparing northern vs. southern regions (Table 1), at urban sites the annual

354 mean concentrations of $NH_3$, $HNO_3$, and $pNH_4^+$ showed smaller non-significant

355 differences (-1~9%), whereas $NO_2$ and $pNO_3^-$ showed larger non-significant increases

356 (34 and 76%, respectively) in the north. By contrast, the mean concentrations of all

357 measured $N_r$ species were significantly ($p<0.05$) higher (by 40-104%) at rural sites in

358 northern region. Similarly, individual concentrations at background sites were 21-71%

359 higher in the northern than southern region. The annual concentrations of total $N_r$ (i.e.,

360 the sum of five $N_r$ species) decreased in the order urban > rural > background in

361 eastern China as a whole and in the north and south regions; further, the annual total

362 $N_r$ concentrations at urban and background sites were 17 and 34% higher ($p>0.05$) in

363 the north than in the south, respectively, whereas those at northern rural sites ($31.6 \pm$

364 $3.8$ µg N m$^{-3}$) were significantly ($p<0.05$) higher than the mean at southern rural sites

365 ($17.0 \pm 1.7$ µg N m$^{-3}$).

366  The monthly VWM concentrations of $NH_4^+$-N, $NO_3^-$-N, and TIN (the sum of

367 $NH_4^+$-N and $NO_3^-$-N) were in the ranges 0.01 (BY)-26.77 (YC), 0.06 (XS)-28.92 (WJ),

368 and 0.09 (XS)-50.29 (YC) mg N L$^{-1}$, respectively (Table S3, Supplement). In eastern

369 China and in each region, the annual VWM concentrations of $NO_3^-$-N and TIN

370 showed a declining trend of urban > rural > background, whereas those of $NH_4^+$-N

371 followed the order rural ≥ urban > background (Table 1). Comparing northern and

372 southern regions, the annual concentrations of $NH_4^+$-N, $NO_3^-$-N, and TIN were



comparable at urban and background sites, and were significantly ($p<0.05$) higher at
northern rural sites.

**3.2 Annual variability in concentrations of $N_r$ species in air and precipitation**

During the 2011-2015 period the annual mean concentrations of measured $N_r$
species in air exhibited no significant trends at the twenty-two selected sites except
for $NH_3$ at four sites (ZZ, DL, ZMD, YL), $HNO_3$ at three sites (DL, LSD, BY),
$pNH_4^+$ at one site (XS), and total $N_r$ at three sites (ZMD, YL, WJ) (Fig. S1a-f,
Supplement). Similarly, no significant trends were found for the annual VWM
concentrations of $NH_4^+$-N, $NO_3^-$-N, and TIN in precipitation at the seventeen selected
sites, with the exception of $NO_3^-$-N at one site (SZ) (Fig. S2a-c, Supplement).
Fig. 2 compares annual average concentrations of all measured $N_r$ species
between the periods 2013-2015 and 2011-2012 for three land use types. In eastern
China the mean concentrations of $NH_3$ and $pNH_4^+$ showed non-significant increases
(10-38%) at all land use types except $pNH_4^+$ at background sites, which showed a
small reduction (8%) (Fig. 2a, d). By contrast, the mean concentrations of remaining
$N_r$ species at three land use types showed smaller and non-significant changes: -8-3%
for $NO_2$ (Fig. 2b), -13-5% for $HNO_3$ (Fig. 2c), and -1-5% for $pNO_3^-$ (Fig. 2e). The
relative changes in the annual total $N_r$ concentration were also not significant, with the
largest increase at rural sites (16%) and smaller increases at urban (4%) and
background (1%) sites (Fig. 2f). Separated by regions, annual mean concentrations of
five $N_r$ species at three land use types mostly showed increases (4-57%) in the north,
and reductions (0.3-21%) in the south (Fig. 2a-f). The relative changes in individual
concentrations at northern rural sites (9% reduction for $HNO_3$, and 9-52% increases
for the other species) and southern rural sites (4% increase for $pNH_4^+$, and 0.3-21%
reductions for other species) were not significant. The annual total $N_r$ concentrations
showed small relative changes (from -1% to 5%) across all land use types in the two
regions, except at northern rural sites, which exhibited a larger but non-significant
increase (25%) (Fig. 2f). Due to significant interannual variability, longer records are
needed to better assess the significance of any concentration changes.





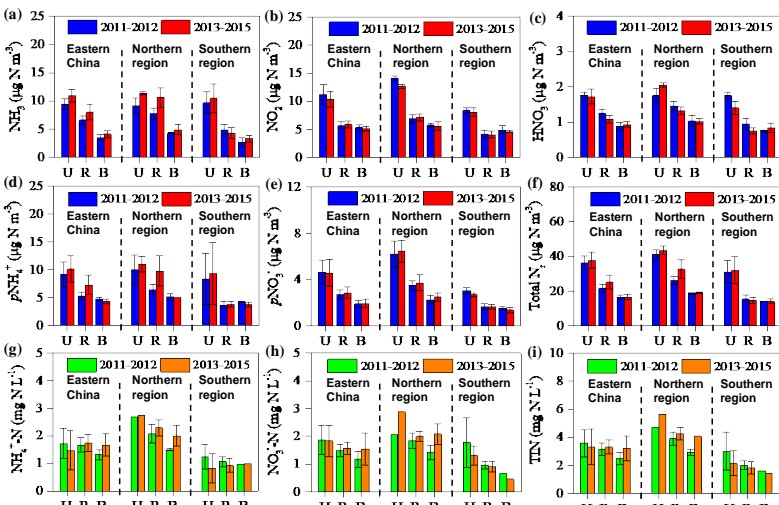


**Figure 2**. Comparison of annual mean concentrations of **(a)** NH$_3$; **(b)** NO$_2$; **(c)** HNO$_3$;

**(d)** $p$NH$_4^+$; **(e)** $p$NO$_3^-$; and **(f)** total N$_r$: sum of all measured N$_r$ in air and

volume-weighted concentrations of NH$_4^+$ **(g)**; NO$_3^-$ **(h)** and total inorganic N (TIN):

sum of NH$_4^+$ and NO$_3^-$ **(i)** in precipitation between the 2011-2012 period and the

2013-2015 period for different land use types in eastern China and its northern and

southern regions. The number of sites for each land use type in each region can be

found in Table S1 in the Supplement. The error bars are the standard errors of means.

In eastern China, the annual VWM concentrations of NH$_4^+$-N, NO$_3^-$-N and TIN

showed the largest increase of 26-31% at background sites, a smaller increase of 4-5%

at rural sites, and a decrease of 2-14% at urban sites; however, those changes were not

significant (Fig. 2g-i). Regionally, their respective concentrations showed increases

(3-45%) in the north and reductions (5-33%) in the south, except for a small increase

(4%) in NH$_4^+$-N at background sites.

**3.3 Seasonal variability in concentrations of N$_r$ species in air and precipitation**

Fig. 3 shows seasonal patterns of NH$_3$, NO$_2$, HNO$_3$, $p$NH$_4^+$, $p$NO$_3^-$ and total N$_r$

concentrations for three land use types in eastern China and its northern and southern

regions, averaged from corresponding measurements at the twenty-seven study sites

(details for each site are given in Tables S4-S9 of the Supplement). Average NH$_3$



concentrations at all land use types decreased in the order summer > spring > autumn >
winter, and significant seasonal differences generally occurred between summer and
winter (Fig. 3a). Conversely, the average $NO_2$ concentration generally showed the
highest value in winter and the lowest in summer; differences between seasonal
concentrations were sometimes significant at rural sites in the south and background
sites, but not at urban sites (Fig. 3b). The seasonal changes in the $HNO_3$ concentration
were generally small and not significant for all land use types (Fig. 3c).
The average $pNH_4^+$ concentration exhibited a non-significant seasonal variation
across all land use types, except for southern rural sites which showed significantly
higher values in winter than in summer (Fig. 3d). The highest $pNH_4^+$ concentrations
mostly occurred in winter. The average $pNO_3^-$ concentrations at all land use types
followed the order winter > spring, ~ autumn > summer; the seasonal changes are
sometimes significant, except for urban sites in eastern China and its northern region
(Fig. 3e). The average concentration of total $N_r$ usually showed small and
non-significant seasonal differences for all land use types (Fig. 3f).

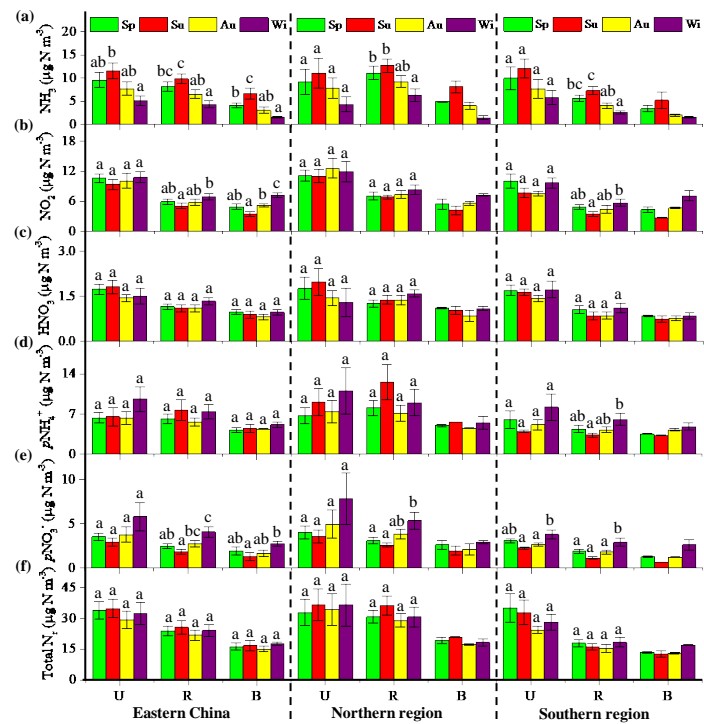




**Figure 3**. Seasonal mean concentrations of **(a)** NH$_3$; **(b)** NO$_2$; **(c)** HNO$_3$; **(d)** $p$NH$_4^+$; **(e)** $p$NO$_3^-$; and **(f)** total N$_r$: sum of all measured N$_r$ in air at different land use types in eastern China and its northern and southern regions. Sp, Su, Au, and Wi represent spring, summer, autumn, and winter, respectively. The number of sites for each land use type in each region can be found in Table S1 in the Supplement. The error bars are the standard errors of means, and different letters on the bars denote significant differences between the sites ($p<0.05$).

In eastern China and its two regions, the seasonal VWM concentrations of NH$_4^+$-N, NO$_3^-$-N and TIN in precipitation at three land use types (averaged from the twenty-seven sites, details in Tables S10-S12 of the Supplement) showed a similar seasonal pattern, with the highest values in winter and the lowest in summer or autumn (Fig. 4a-c). Significant seasonal differences usually occurred between winter and the other three seasons at all land use types, except background sites and southern urban sites.

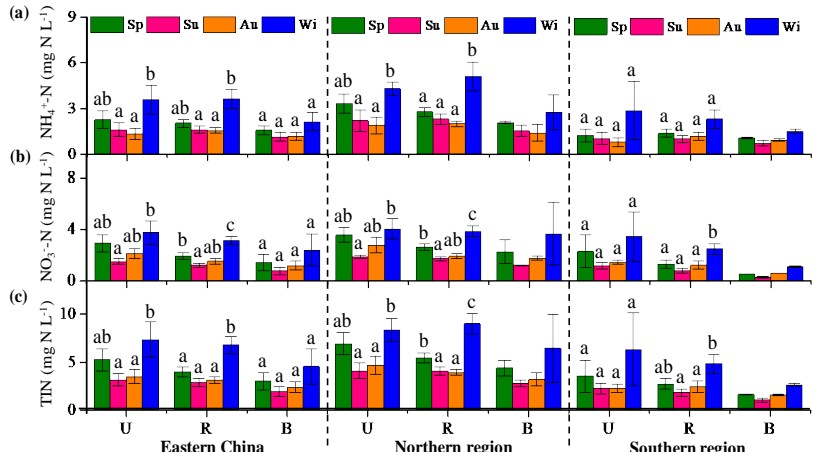

**Figure 4**. Seasonal mean concentrations of NH$_4^+$ **(a)**; NO$_3^-$ **(b)** and total inorganic N (TIN): sum of NH$_4^+$ and NO$_3^-$ **(c)** in precipitation at different land use types in eastern China and its northern and southern regions. Sp, Su, Au, and Wi represent spring, summer, autumn, and winter, respectively. The number of sites for each land use type in each region can be found in Table S1 in the Supplement. The





error bars are the standard errors of means, and different letters on the bars denote

significant differences between the sites ($p<0.05$).

**3.4 Spatial variability in dry and wet/bulk N deposition of $N_r$ species**

Dry deposition fluxes of $NH_3$, $HNO_3$, $NO_2$, $pNH_4^+$, and $pNO_3^-$ ranked in the

order urban > rural > background in eastern China and in both southern and northern
regions (except for $pNH_4^+$ in the north) (Table 2). Comparing northern and southern
regions, at urban sites the mean dry $pNH_4^+$ deposition was slightly higher (2%) in the
north, whereas larger enhancements (24-69%) in the mean fluxes were found in the
north for the remaining $N_r$ species. By contrast, individual fluxes were significantly
higher (by 64-138%) at northern rural sites, except for $HNO_3$ which showed a large
non-significant increase (58%). At northern background sites, the mean dry deposition
fluxes of $NH_3$ and $NO_2$ were much higher (159%) and lower (68%), respectively;
however, only small differences in the means were found for $HNO_3$ (6% lower in the
north), $pNH_4^+$ (5% lower), and $pNO_3^-$ (14% higher). The spatial pattern of total N dry
deposition flux (the sum of the fluxes of the five $N_r$ species) by land use types ranked
in the same order as individual $N_r$ species in eastern China. Compared with the
southern region, mean total N fluxes in the north region were significantly higher (by
85%) at rural sites, but showed non-significant increases at urban and background
sites (33 and 38%, respectively).

The wet/bulk deposition fluxes of $NH_4^+$-N, $NO_3^-$-N, and TIN ranked in the order

urban > rural > background in eastern China and in each region (except for $NH_4^+$-N in
the south) (Table 2). In addition, their respective fluxes were generally comparable in
northern and southern regions.











**Table 2.** Annual average (standard error) dry and wet/bulk deposition fluxes (kg N ha$^{-1}$ yr$^{-1}$) of various N$_r$ compounds at different land use types in eastern China and its northern and southern regions for the 5-year period 2011-2015.

| Region[a] | LUY[b] | Dry deposition | | | | | | Wet/bulk deposition | | |
|---|---|---|---|---|---|---|---|---|---|---|
| | | NH$_3$ | NO$_2$ | HNO$_3$ | $p$NH$_4^+$ | $p$NO$_3^-$ | Total N$_r$ | NH$_4^+$ | NO$_3^-$ | TIN |
| EC | Urban | 12.6 | 4.4 | 7.7 | 4.8 | 2.1 | 31.7 | 12.6 | 15.4 | 28.0 |
| | (n=6) | (1.4) | (1.2) | (1.6) | (1.4) | (0.5) | (4.6) | (1.9) | (0.7) | (2.2) |
| | Rural | 9.1 | 2.9 | 4.6 | 4.0 | 1.5 | 22.1 | 11.9 | 10.2 | 22.1 |
| | (n=17) | (0.9) | (0.3) | (0.6) | (0.7) | (0.2) | (2.3) | (1.0) | (0.5) | (1.4) |
| | BKD[c] | 7.9 | 1.8 | 3.5 | 1.9 | 0.8 | 15.8 | 10.7 | 7.7 | 18.4 |
| | (n=4) | (2.1) | (0.6) | (0.2) | (0.3) | (0.1) | (1.5) | (1.8) | (0.3) | (1.8) |
| NREC | Urban | 13.9 | 5.2 | 9.4 | 4.9 | 2.7 | 36.2 | 13.9 | 14.1 | 28.0 |
| | (n=3) | (1.9) | (2.5) | (3.0) | (1.9) | (1.0) | (8.2) | (3.5) | (1.0) | (4.4) |
| | Rural | 12.1** | 3.6* | 5.7 | 5.7* | 2.1** | 29.3** | 12.3 | 10.3 | 22.6 |
| | (n=8) | (1.3) | (0.4) | (1.0) | (1.2) | (0.3) | (3.2) | (1.3) | (0.7) | (1.8) |
| | BKD | 11.4 | 0.9 | 3.4 | 1.9 | 0.8 | 18.4 | 7.8 | 7.6 | 15.4 |
| | (n=2) | (0.6) | (0.7) | (0.3) | (0.7) | (0.2) | (0.7) | (1.4) | (0.8) | (0.6) |
| SREC | Urban | 11.2 | 3.6 | 5.9 | 4.8 | 1.6 | 27.2 | 11.4 | 16.6 | 28.0 |
| | (n=3) | (2.0) | (0.3) | (0.6) | (2.6) | (0.2) | (4.0) | (2.0) | (0.4) | (2.1) |
| | Rural | 6.5 | 2.2 | 3.6 | 2.4 | 1.0 | 15.8 | 11.6 | 10.2 | 21.8 |
| | (n=9) | (0.5) | (0.4) | (0.6) | (0.4) | (0.2) | (1.4) | (1.5) | (0.9) | (2.2) |
| | BKD | 4.4 | 2.7 | 3.6 | 2.0 | 0.7 | 13.3 | 13.6 | 7.9 | 21.5 |
| | (n=2) | (1.0) | (0.2) | (0.3) | (0.1) | (0.1) | (0.7) | (0.1) | (0.1) | (0.1) |

[a] EC: eastern China; NREC: northern region of eastern China; SREC: southern region of eastern China. [b] LSY: land use type; n denotes number of monitoring sites. [c] BKD: Background. *Significant at the 0.05 probability level. **Significant at the 0.01 probability level.

### 3.5 Annual variability in dry and wet/bulk N deposition

The annual trends of dry deposition fluxes of individual N$_r$ species at the twenty-two selected sites are consistent with trends in their respective ambient concentrations, except for HNO$_3$ at three sites (SZ, LSD, and ZY) (Figs. S3a-e and S1a-e, Supplement). A consistent picture is also seen for the total dry N deposition fluxes at all but two sites (DL and WJ) (Figs. S3f and S1f, Supplement). Similarly, the annual trends of wet/bulk deposition fluxes of NH$_4^+$-N, NO$_3^-$-N and TIN at seventeen selected sites are similar to their respective concentrations in precipitation (Fig. S4a-c,





Supplement).

In eastern China the annual average dry deposition fluxes of $NH_3$, $NO_2$, $HNO_3$,

$pNH_4^+$ and $pNO_3^-$ showed non-significant increases (2-39%) or reductions (1-19%)
between the periods 2011-2012 and 2013-2015 at the three land use types (Fig. 5a-e),
similar in sign and magnitude to their respective concentrations described earlier. The
annual average total N dry deposition fluxes showed small and non-significant
increases across the study periods: 2% at urban sites, 9% at rural sites, and 7% at
background sites (Fig. 5f). The sign and magnitude of period-to-period changes in dry
deposition and ambient concentrations of all measured $N_r$ species were generally
similar between the southern and northern regions.

Wet/bulk deposition fluxes of $NH_4^+$-N, $NO_3^-$-N, and TIN generally decreased

(4-29%) between 2011-2012 and 2013-2015 periods at all land use types in eastern
China; one exception was $NO_3^-$-N, which exhibited a small increase (3%) at urban
sites (Fig. 5g-i). Similar tendencies were also observed in both northern and southern
regions.

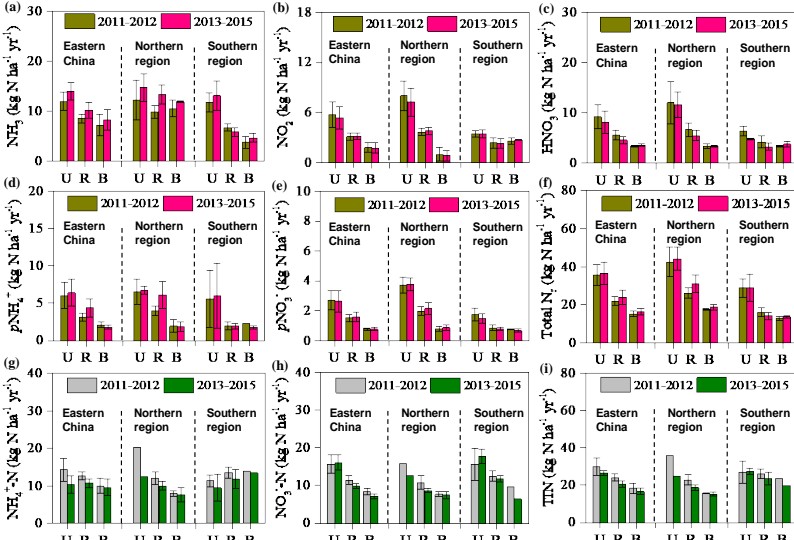


**Figure 5**. Comparison of dry deposition of **(a)** $NH_3$; **(b)** $NO_2$; **(c)** $HNO_3$; **(d)** $pNH_4^+$;
**(e)** $pNO_3^-$; and **(f)** total $N_r$: sum of all measured $N_r$ in air and wet/bulk deposition of
$NH_4^+$ **(g)**; $NO_3^-$ **(h)** and total inorganic N (TIN): sum of $NH_4^+$ and $NO_3^-$ **(i)** in





precipitation between the 2011-2012 period and the 2013-2015 period for different
land use types in eastern China and its northern and southern regions. The number of
sites for each land use type in each region can be found in Table S1 in the Supplement.

The error bars are the standard errors of means.


**3.6 Seasonal variability in dry and wet/bulk deposition of $N_r$ species**
Seasonal variations of dry deposition of individual $N_r$ species at each site are
shown in Tables S4-S9 in the Supplement. In eastern China and in each region, dry
$NH_3$ deposition fluxes at all land use types followed the order summer > spring >
autumn > winter, with the seasonal changes usually significantly different (Fig. 6a).
Similarly, dry the $NO_2$ deposition flux was also at its minimum in winter, but its
maximum was found in summer at urban and rural sites and in autumn at background
site; seasonal differences in most cases were not significant (Fig. 6b). Seasonal
patterns of dry $HNO_3$ deposition flux at all land use types were similar to those for dry
$NH_3$ deposition fluxes, and the resulting seasonal changes were sometimes significant,
except at northern urban sites (Fig. 6c).

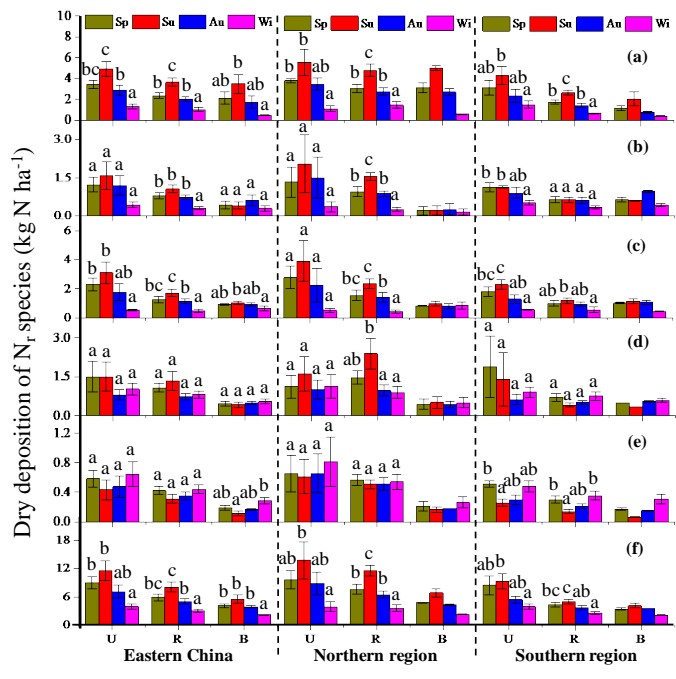




**Figure 6**. Seasonal mean dry deposition of **(a)** $NH_3$; **(b)** $NO_2$; **(c)** $HNO_3$; **(d)** $pNH_4^+$;
**(e)** $pNO_3^-$; and **(f)** total $N_r$: sum of all measured $N_r$ in air at different land use types in
eastern China and its northern and southern regions. Sp, Su, Au, and Wi represent
spring, summer, autumn, and winter, respectively. The number of sites for each land
use type in each region can be found in Table S1 in the Supplement. The error bars are
the standard errors of means, and different letters on the bars denote significant
differences between the sites ($p<0.05$).

Dry $pNH_4^+$ deposition fluxes peaked in spring or summer at urban and rural sites,
but remained at similar levels across the four seasons at background sites; however,
no significant seasonal variations were found at any land use types except for rural
sites in the north (Fig. 6d). Dry $pNO_3^-$ deposition fluxes were higher in spring and
winter than in summer and autumn at all land use types, and the seasonal changes
were sometimes significant at background sites and at southern urban and rural sites
(Fig. 6e). Total dry N deposition fluxes at all land use types showed similar seasonal
variations to dry $NH_3$ deposition, with the highest values in summer and the lowest in
winter; significant seasonal differences generally were observed between winter and
the other three seasons (Fig. 6f).
Wet/bulk deposition fluxes of $NH_4^+$-N, $NO_3^-$-N, and TIN all showed significant
seasonal variation at urban and rural sites, but not at background sites, with the
highest values in summer and the lowest in winter (Fig. 7a-c).



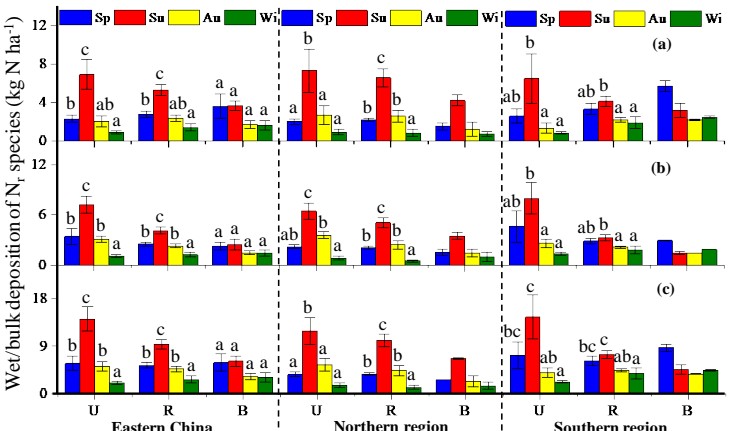


**Figure 7**. Seasonal mean wet/bulk deposition of $NH_4^+$ **(a)**; $NO_3^-$ **(b)** and total

inorganic N (TIN): sum of $NH_4^+$ and $NO_3^-$ **(c)** in precipitation at different land use

types in eastern China and its northern and southern regions. Sp, Su, Au, and Wi

represent spring, summer, autumn, and winter, respectively. The number of sites for

each land use type in each region can be found in Table S1 in the Supplement. The

error bars are the standard errors of means, and different letters on the bars denote

significant differences between the sites ($p<0.05$).

**3.7 Spatial-temporal variability in total annual dry and wet/bulk deposition of $N_r$**

**species**

  In eastern China total annual mean N deposition (dry plus wet/bulk) fluxes at

rural and background sites were comparable (on average, 44.3 ± 3.0 and 34.3 ± 0.7 kg

N ha$^{-1}$ yr$^{-1}$, respectively), but significantly lower than those at urban sites (59.7 ± 6.1

kg N ha$^{-1}$ yr$^{-1}$) (Tables 1 and 2, and Fig. S5, Supplement). Similar tendencies for total

N deposition fluxes were observed in the southern region, while in the north a

significant difference was only found between urban and background sites (Fig. S5,

Supplement). From 2011 to 2015, no significant annual trend was found in the total N

deposition at sixteen selected sites (Fig. S6a, Supplement). The total annual mean N

deposition fluxes at three land use types showed small and non-significant reductions

(1-5%) between 2011-12 and 2013-15 (Fig. S6b, Supplement). Regionally, the total




fluxes at each land use type were of similar magnitude in the two periods. Also, the $NH_x$ (wet/bulk $NH_4^+$-N deposition plus dry deposition of $NH_3$ and particulate $NH_4^+$)/$NO_y$ (wet/bulk $NO_3^-$-N deposition plus dry deposition of $NO_2$, $HNO_3$ and particulate $NO_3^-$) ratio showed a non-significant annual trend across all sites (**Fig. 8a**). At all land use types, the averaged ratios were slightly higher in the 2013-2015 period than in the 2011-2012 period (Fig. 8b).

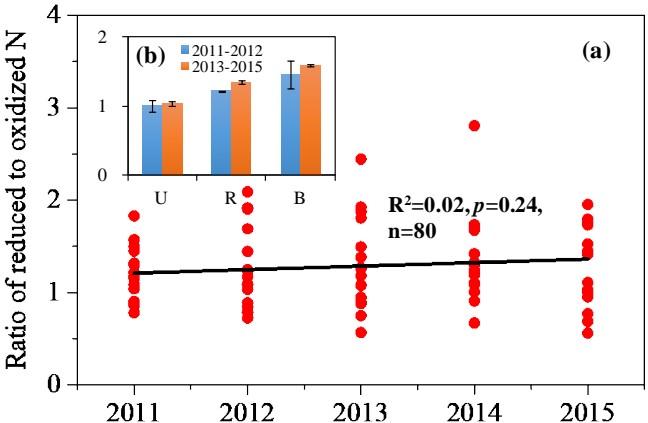

**Figure 8**. Annual trend of the ratio of $NH_x$ (wet/bulk $NH_4^+$-N deposition plus dry deposition of $NH_3$ and particulate $NH_4^+$) to $NO_y$ (wet/bulk $NO_3^-$-N deposition plus dry deposition of $NO_2$, $HNO_3$ and particulate $NO_3^-$) across sixteen selected sites **(a)**, with a comparison between the 2011-2012 period and the 2013-2015 period for different land use types in eastern China **(b)**. The number of sites with the same land use type can be found in Fig. S6 in the Supplement.

## 4. Discussion

### 4.1 Comparisons of $NH_3$ and $NO_2$ measurements with satellite data

Eastern China, as a highly industrialized and polluted region and has been proven to be a hot spot of $N_r$ ($NH_3$ and $NO_x$) emission and deposition globally (Vet et al., 2014; Kanakidou et al., 2016). The results presented above showed that, in eastern China, annual mean concentrations of measured $N_r$ species in air and precipitation were generally higher in the north than in the south (Table 1). This is likely due to



higher consumption of energy and application of N-fertilizers, along with lower
precipitation amounts in the north, previously identified as key factors affecting
spatial patterns of N deposition in China (Jia et al., 2014; Zhu et al., 2015). Because
only 27 sites covering a range of land use types were included in the present study,
additional information would be valuable in determining whether the observed spatial
patterns adequately represent conditions in eastern China. To address this issue, we
use measured $NH_3$ and $NO_2$ concentrations to evaluate remote sensing techniques for
retrieving $NH_3$ and $NO_2$ concentrations. If accurate, those remote sensing techniques
are well suited to ascertain regional species distributions. $NH_3$ and $NO_x$ are primary
emissions with important anthropogenic emissions (Fowler et al., 2013).   NO, the
main component of emitted $NO_x$, is oxidized in the atmosphere to $NO_2$. $NO_2$ is further
oxidized via daytime or nighttime chemistry to $HNO_3$ (Khoder, 2002). $NH_3$ and
$HNO_3$ can react to form fine particle ammonium nitrate (Seinfeld and Pandis, 2006).
Thus, spatial patterns of $NH_3$ and $NO_2$ observed from space can be useful indicators
of reduced and oxidized $N_r$ pollution over eastern China.

From satellite observations (Fig. 9a, b), it can be seen that both IASI_$NH_3$ and

OMI_$NO_2$ columns show clearly higher values over the northern region of eastern
China. Overall, satellite observations and surface measurements for $NH_3$ and $NO_2$
(plotted on the maps of Fig. 9a, b) show a similar spatial pattern. Significant positive
correlations were found between IASI_$NH_3$ column observations and NNDMN_$NH_3$
measurements (r=0.72, $p$<0.001) (Fig. 9c) and between OMI_$NO_2$ observations and
NNDMN_$NO_2$ measurements (r=0.86, $p$<0.001) (Fig. 9d) at the 27 surface
measurement locations, suggesting that satellite measurements of $NH_3$ and $NO_2$ can
be used to capture regional differences in $NH_3$ and $NO_2$ pollution. Looking beyond
the surface measurement location, the satellite observations further confirm the
existence of greater $N_r$ pollution in the northern region of eastern China than in the
southern region.



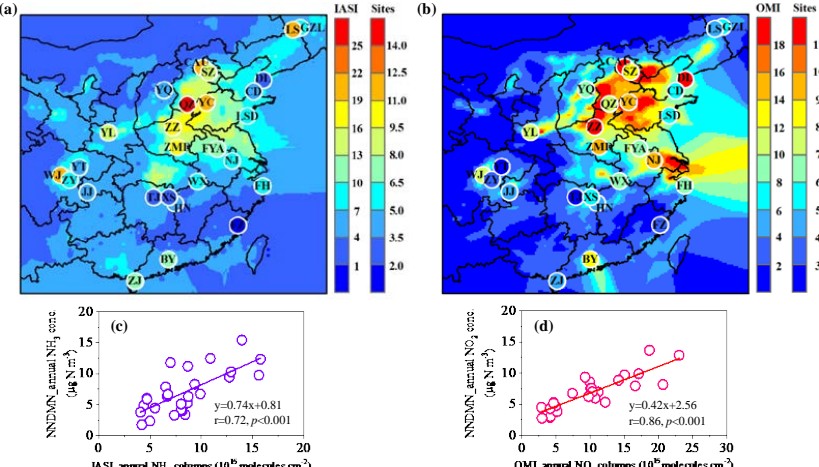

**Figure 9**. Spatial variation of atmospheric $N_r$ in eastern China: **(a)**
NNDMN_NH$_3$ concentrations vs. IASI_NH$_3$ columns; **(b)** NNDMN_NO$_2$
concentrations vs. OMI_NO$_2$ columns; **(c)** relationship of NNDMN_NH$_3$
concentrations vs. IASI_NH$_3$ columns; **(d)** relationship of NNDMN_NO$_2$
concentrations vs. OMI_NO$_2$ columns.

To further explore temporal concentration variability, monthly mean satellite
NH$_3$ and NO$_2$ columns are compared with monthly mean ground concentrations of
NH$_3$ and NO$_2$ (Figs. S7 and S8, Supplement). The linear correlation between satellite
columns and surface NH$_3$ concentrations is significant ($p<0.05$) at the ten sites
($r=0.32$-$0.87$) in the northern region and at four sites ($r=0.46$-$0.84$) in the southern
region (Fig. S7, Supplement), while the linear correlation between satellite columns
and surface NO$_2$ concentrations is significant at the ten sites ($r=0.28$-$0.68$) in the
northern region and nine sites ($r=0.36$-$0.66$) in the southern region (Fig. S8,
Supplement). These results indicate that the OMI_NO$_2$ retrieval can well capture the
temporal variations of surface NO$_2$ concentrations over eastern China, whereas the
IASI_NH$_3$ retrievals better capture temporal variability in surface concentrations for
the northern region. The weak correlations observed between IASI_NH$_3$ observations
and surface measurements at ten of the fourteen sites in the southern region (Fig. S7,
Supplement) suggest that the IASI_NH$_3$ observations need to be improved for
investigating temporal variability in NH$_3$ concentration, despite that the satellite



observation is at a specific time of day while the surface concentrations integrate
across the diurnal cycle of emissions and mixing layer evolution. It should be noted
that a direct comparison between surface concentration and satellite column
measurements is inevitably affected by many factors, such as changes in boundary
layer height, vertical profiles of species, and interferences from cloud and aerosol
(Van Damme et al., 2015). Nevertheless, the ratio of satellite column to surface
concentration measurements is meaningful as it can provide insight into sensitivity of
a satellite retrieval to variation in the concentration of a gas in the surface layer (Meng
et al., 2008). To make a more accurate comparison, the vertical profile is
recommended to convert the columns to the ground concentrations in future work.
**4.2 Seasonal variations of $N_r$ concentration and deposition**
The seasonal concentrations of $N_r$ species in air and precipitation are dependent
on their sources and meteorological conditions. The highest concentrations of $NH_3$ in
summer at all land use types (Fig. 3a) are most likely due to enhanced $NH_3$ emission
from natural and fertilized soils, and biological sources such as humans, sewage
systems and organic waste in garbage containers (Chang et al., 2016). Zhang et al.
(2018) showed that $NH_3$ emissions in China show a strong summer peak, with
emissions about 50% higher in summer than spring and autumn. The lowest
concentrations of $NH_3$ in winter (Fig. 3a) can be ascribed to the reduced $NH_3$
volatilization at low air temperature, high snow coverage, and low agricultural
activities (Cao et al., 2009) as well as consumption of $NH_3$ to form $NH_4NO_3$ (Fig. 3a,
d and e) and/or $(NH_4)_2SO_4$. The lower $NO_2$ concentration in summer (Fig. 3b) might
result from greater atmospheric mixing in a deeper boundary layer and a higher rate of
oxidation of $NO_2$ to $HNO_3$ by reaction with OH (Atkins and Lee, 1995), which is
more abundant in summer due to greater photochemical activity. Increased $NO_2$
emissions from greater coal combustion for domestic heating (from middle November
to middle March) in Northern China may also enhance $NO_x$ emissions and subsequent
$NO_2$ concentrations in autumn/winter (Zhao et al., 2011).
Nitric acid is a secondary pollutant, formed through gas phase reaction of $NO_2$
with the OH radical, reaction of $NO_3$ with aldehydes or hydrocarbons or hydrolysis of



$N_2O_5$ (Khoder, 2002). Nitric acid concentrations are expected to be further influenced
by air temperature, relative humidity and ambient $NH_3$ concentrations (Allen et al.,
1989); fine particle $NH_4NO_3$ formation is favored at low temperatures and high
relative humidities. Due to a lack of information regarding primary formation
pathways and influencing factors at our study sites, we cannot offer a definitive
explanation for small and differing seasonal patterns of $HNO_3$ concentrations
observed at the three land use types (Fig. 3c). Particulate $NH_4^+$ and $NO_3^-$ are also
mainly generated via chemical reactions between $NH_3$ and inorganic acids (e.g.,
$HNO_3$, $H_2SO_4$). We found that concentrations of $pNH_4^+$ and $pNO_3^-$ at all land use
types usually peaked in winter (Fig. 3e, f). Low temperature and high emissions of
$NO_x$ and $SO_2$ in winter are favorable for formation of ammonium sulfate $((NH_4)_2SO_4)$
and ammonium nitrate $(NH_4NO_3)$ aerosols (Xu et al., 2016), consistent with higher
concentrations of $pNH_4^+$ and $pNO_3^-$. In addition, in winter temperature inversions in
combination with stable meteorological conditions (e.g., low wind speed) limit
horizontal and vertical exchange of pollutants, and further elevated atmospheric
$pNH_4^+$ and $pNO_3^-$ levels (Liu et al., 2017).
Ammonium-N and nitrate-N in precipitation mainly originate from
corresponding reduced (e.g., $NH_3$, $pNH_4^+$) and oxidized (e.g., $HNO_3$, $NO_2$, $pNO_3^-$) N
in air, scavenged respectively, by rain and/or snow events (Seinfeld and Pandis, 2006).
At all land use types, the seasonal variation of $NH_4^+$-N concentration in precipitation
was opposite to that of reduced N (the sum of $NH_3$ and $pNH_4^+$) concentrations (Figs.
4a and S9a in the Supplement), whereas a similar seasonal pattern was found between
$NO_3^-$-N and oxidized N (the sum of $HNO_3$, $NO_2$ and $pNO_3^-$) concentrations (Figs. 4b
and S9b in the Supplement). Higher precipitation amounts in summer could account
for lower $NH_4^+$-N concentrations in summer (Figs. 4a and S10 in the Supplement) due
to a dilution effect (Xu et al., 2015). In contrast, seasonal variations of rainwater
$NO_3^-$-N concentrations were more likely dominated by seasonal changes in oxidized
N concentrations rather than precipitation amount.
The seasonal variation of $NH_3$ dry deposition is generally similar to that of $NH_3$
concentration (Figs. 3a and 6a). Given comparable seasonal mean $V_d$ for $NH_3$ across



the four seasons in most cases (Fig. S11a-c, Supplement), the seasonality of $NH_3$
deposition is mainly dominated by changes in ambient $NH_3$ concentrations. Seasonal
deposition fluxes of $NO_2$ and $HNO_3$ both differ appreciably (Fig. 6b, c), showing
similar variation to seasonality of their respective $V_d$ values (Fig. S11d-i, Supplement).
Given weaker seasonal fluctuations of $NO_2$ and $HNO_3$ concentrations, the seasonality
of $NO_2$ and $HNO_3$ dry deposition are primarily functions of changes in $V_d$. Similar
analyses suggest that seasonal variation of $pNO_3^-$ dry deposition was mainly caused
by differences in seasonal $pNO_3^-$ concentrations (Figs. 3e and 6e), whereas that of
$pNH_4^+$ dry deposition was primarily driven by seasonal changes in $V_d$ (Figs. 6c and
S11j-l, Supplement).

**4.3 The role of $NH_3$ in mitigation of $N_r$ air pollution**

The latest pollutant emissions statistics from the Chinese Ministry of
Environmental                                      Protection
(http://www.zhb.gov.cn/gkml/hbb/qt/201507/t20150722_307020.htm) showed that
total annual emissions of $SO_2$ and $NO_x$ were reduced by 12.9% and 8.6% in 2014,
respectively, compared with those in 2010. This suggests that the goal set for the 12[th]
FYP period was fulfilled ahead of time. Our field measurements demonstrate that
annual mean concentrations of each $N_r$ species and total $N_r$ did not show significant
decreasing trends at most sites during the 2011-2015 period (Fig. S1a-f, Supplement).
Furthermore, annual mean total $N_r$ concentrations showed non-significant increases
(1-16%) at three land use types during the 2013-2015 period compared with
2011-2012 (Fig. 2f). These results together suggest that $N_r$ pollution may be not
effectively mitigated in eastern China during the 12[th] FYP, likely due to the absence of
$NH_3$ regulations, despite enforcement of a "Zero Increase Action Plan" by the
Ministry of Agriculture for national fertilizer use (X. J. Liu et al., 2016).
Ammonia is the primary alkaline gas in the atmosphere. It plays an important
role in formation of $(NH_4)_2SO_4$ and $NH_4NO_3$ aerosols (Seinfeld and Pandis, 2006).
These secondary inorganic aerosols account for 40–57 % of the $PM_{2.5}$ concentrations
in eastern China (Yang et al., 2011; Huang et al., 2014). Based on monthly mean
molar concentrations, there were significant positive linear correlations between $NH_3$





and $p$NH$_4^+$, NO$_2$ and $p$NO$_3^-$, SO$_2$ and $p$SO$_4^{2-}$, $p$NH$_4^+$ and $p$NO$_3^-$, and $p$NH$_4^+$ and
$p$SO$_4^{2-}$ at all land use land types except for a non-significant relationship of NH$_3$ with
$p$NH$_4^+$ at background sites (Fig. 10a-e). These results suggest that the precursor gases
are responsible for the formation of secondary inorganic ions (i.e., $p$NH$_4^+$, $p$NO$_3^-$, and
$p$SO$_4^{2-}$) locally at urban and rural sites, while secondary inorganic ions at background
sites likely originated from long-distance transport. The ratio of NH$_3$ to NH$_x$ (NH$_3$
plus $p$NH$_4^+$) concentrations at urban (0.53 ± 0.15) and rural (0.52 ± 0.16) sites
exceeded values at background (0.43 ± 0.16) sites. According to Walker et al. (2004),
a value greater than 0.5 indicates that NH$_x$ is more likely to be from local sources as
opposed to long-range transport.

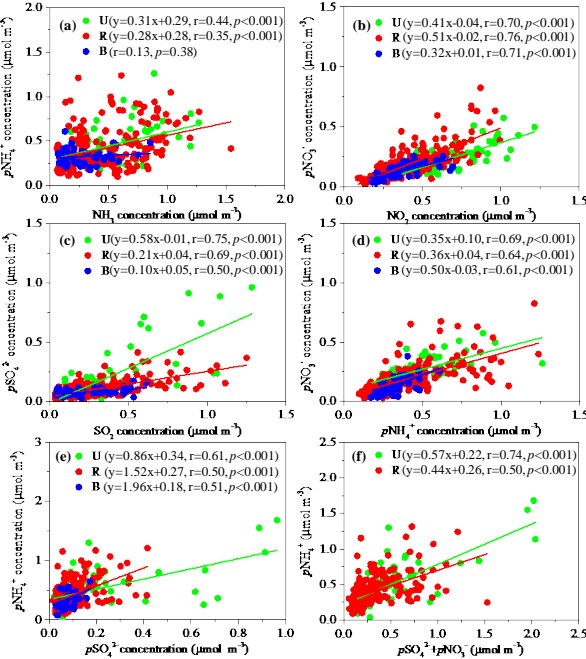


**Figure 10**. Correlations of monthly mean molar concentrations of (a) $p$NH$_4^+$ vs. NH$_3$;
(b) $p$NO$_3^-$ vs. NO$_2$; (c) $p$SO$_4^{2-}$ vs. SO$_2$; (d) $p$NO$_3^-$ vs. $p$NH$_4^+$; (e) $p$NH$_4^+$ vs. $p$SO$_4^{2-}$; (f)
$p$NH$_4^+$ vs. ($p$SO$_4^{2-}$ + $p$NO$_3^-$) at three land use types in eastern China. The number of
sites with the same land use type in each region can be found in Table S1 in the
Supplement.
It is known that NH$_3$ in the atmosphere is preferentially neutralized by H$_2$SO$_4$ to



form (NH$_4$)$_2$SO$_4$ and/or NH$_4$HSO$_4$, with any remainder available for potential reaction
with HNO$_3$ to form NH$_4$NO$_3$. At urban and rural sites, monthly mean $p$NH$_4^+$
concentrations significantly positively correlated with the sum of $p$SO$_4^{2-}$ and $p$NO$_3^-$
concentrations (Fig. 10f). However, the slopes of regression equations between them
were both smaller than unity (0.57 and 0.44 at urban and rural sites, respectively),
indicating an incomplete neutralization of acidic species (HNO$_3$ and H$_2$SO$_4$) by NH$_3$
at urban and rural sites. In other words, NH$_3$ is a factor limiting the formation of
secondary inorganic ions. A model simulation by Wang et al. (2011) found that,
without NH$_3$ emission controls, NO$_3^-$ in PM$_{2.5}$ will be enhanced by 10% in 2030
compared with 2005 in China, despite improved NO$_x$ emissions controls. As reported
by Zhang et al. (2017), total NH$_3$ emissions in China increased from 12.1 Tg N yr$^{-1}$ in
2000 to 15.6 Tg N yr$^{-1}$ in 2015 at an annual rate of 1.9%. In contrast, total emissions
of NO$_x$ and SO$_2$ have decreased or stabilized in recent years, and were estimated to be
8.4 Tg N yr$^{-1}$ and 12.5 Tg S yr$^{-1}$ in 2014, respectively (Xia et al., 2016). Based on
these factors, implementation of NH$_3$ control strategies, relative to current NO$_x$ and
SO$_2$ emission controls, should be considered to mitigate atmospheric N$_r$ pollution.
**4.4 The role of NH$_3$ emission in control of N deposition**
The present results showed that total dry N deposition fluxes at three land use
types were higher in the northern region of eastern China than in the southern region
(Table 1), mainly due to higher NH$_3$ dry deposition resulting from higher NH$_3$
concentrations in the north. This is especially true for northern rural sites (Table 1),
mostly located in the North China Plain (NCP) (see details in Xu et al. (2015)). The
NCP (that is, Beijing and Tianjin cities and Hebei, Henan, and Shandong provinces), a
highly populated region with intensive agricultural production, contributes 30-40% of
the total annual NH$_3$ emissions in China (Huang et al., 2012). Thus, we anticipate that
reducing NH$_3$ emissions can effectively control N deposition.
To further examine contributions of NH$_3$ emissions to total (wet plus dry) N
deposition at each site and over eastern China, we conducted model sensitivity tests
using the nested GEOS-Chem atmospheric chemistry model driven by the GEOS-5
assimilated meteorological fields at a horizontal resolution of $1/2^o \times 2/3^o$. The model





used anthropogenic emissions from the Multi-Resolution Emission Inventory of
China (MEIC, http://meicmodel.org) for the year 2010, except for $NH_3$ emissions that
are taken from the Regional Emission in Asia (REAS-v2) inventory (Kurokawa et al.,
2013), with an improved seasonality derived by Zhao et al. (2015). In brief,
anthropogenic sources of $NH_3$ emissions include fertilizer use, livestock, human waste,
and fuel combustion (that in power plant, industry, transportation and residential),
whereas $NO_x$ emission sources include industry, power, transportation, and residential.
Both $NH_3$ and $NO_x$ have natural sources (including lighting, biomass burning and soil
emissions). It should be pointed out that fertilizer $NH_3$ emissions include both
chemical fertilizer and manure fertilizer. Details of the model emissions and
mechanisms have been described elsewhere (Zhao et al., 2017, Xu et al., 2018).
We evaluate the model simulations by comparing with measured bulk (both
$NH_4^+$-N and $NO_3^-$-N) fluxes. The model biases for bulk $NH_4^+$-N and $NO_3^-$-N
deposition were 23 and -23%, respectively (Fig. S8, Supplement). These biases are
reasonable, given uncertainties in $N_r$ emissions and predictions of meteorology. Given
that model evaluation is not central to this work, we presented the details in Sect. S1
in the Supplement. As shown in Fig. 11, fertilizer use is the dominant source of total
N deposition at all sites, with contributions between 16-50%. Also, over eastern China
the largest contribution was from fertilizer use (36%) relative to livestock (10%),
industry (14%), power plant (11%), transportation (9%), and other sources (20%, the
sum of contributions from human waste, residential activities, soil, lighting and
biomass burning). These results indicate that reducing $NH_3$ emissions from improper
fertilizer (including chemical and organic fertilizer) application should be a priority in
curbing N deposition in eastern China.



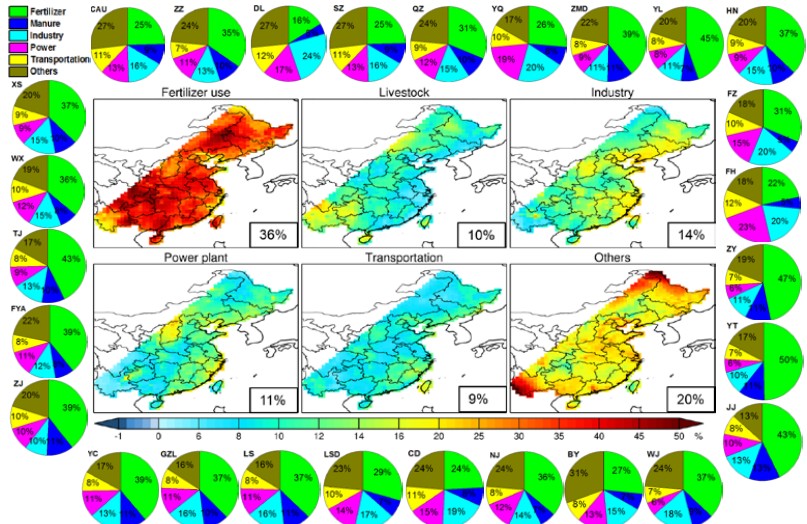


**Figure 11**. Fractional contributions to total N deposition from emission sectors (i.e.
fertilizer use, livestock, industry, power plant, transportation, and others including
emissions from human waste, residential activities, soil, lighting and biomass burning)
at the twenty-seven sites and over eastern China.

**4.5 Deposition response to emission change**
Similar to $N_r$ concentrations, there were no significant decreasing trends in dry
and bulk deposition of total N or of individual $N_r$ species at almost all study sites
(Figs. S3 and S4, Supplement). In addition, we found that changes in annual mean
deposition fluxes of various $N_r$ species are fairly small between the 2013-2015 and
2011-2012 periods (Fig. 5). These results suggest that current emission controls did
not effectively reduce N deposition in eastern China.
To further assess the relationship between emission and deposition change, we
considered the emissions of $SO_2$, $NO_x$ and $NH_3$ affecting the sixteen study sites with
continuous and simultaneous dry and bulk deposition measurements (Fig. S6 and
Table S1, Supplement). The regional $NH_3$ emission data for 2011-2015 were derived
from Zhang et al. (2017), while $SO_2$ and $NO_x$ emission data for 2011-2014 were
derived from Xia et al. (2016) (emission data for the year 2015 were provided by Prof.
Yu Zhao, and were unpublished). We compared these annual data with annual mean



deposition values from the 16 sites. It should be noted that such assessment is subject
to some uncertainty, as emission data was estimated based on the areas belonging to
eastern China.

A clear decreasing trend in $SO_2$ and $NO_x$ emissions was observed, with

reductions of 32% and 25% in 2015 compared to 2011, respectively (Fig. 12a, b). This
reduction is directly related to the widespread use of selective catalytic reduction and
flue gas de-sulfurization on power plants and industries (Van der A et al., 2017), and
to a lesser extent to the introduction of new emission standards for cars (F. Liu et al.,
2016). In contrast, $NH_3$ emissions generally showed a gradual increasing trend
between 2011 and 2015 (Fig. 12c), as control strategies have not yet been enacted and
implemented for $NH_3$ emissions in China.

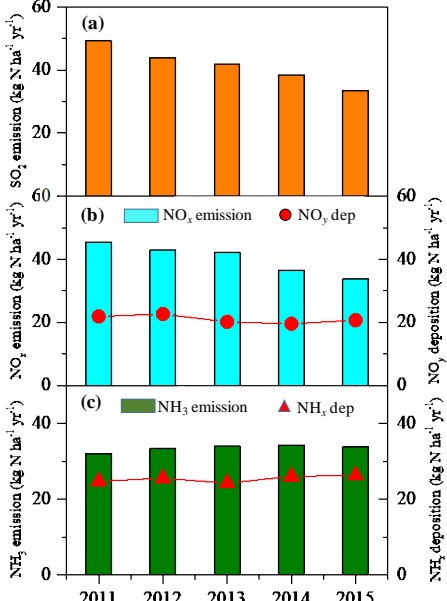


**Figure 12**. Emission of $SO_2$ (a), $NO_x$ (b) and $NH_3$ (c) obtained as average data from
the areas belonging to eastern China, compared with deposition values in the same
periods (mean values from the sixteen sites showing in Fig. S6 and Table S1 in the
Supplement , 5-year averages).

Regarding N deposition, a non-significant increasing trend was found for $NH_x$



(slope=0.36 kg N ha$^{-1}$ yr$^{-1}$) between the 2011 and 2015 period, whereas NO$_y$
deposition exhibited a non-significant decreasing trend (slope=0.54 kg N ha$^{-1}$ yr$^{-1}$).
Also, there were non-significant linear correlations between NH$_x$ deposition and NH$_3$
emission and between NO$_y$ deposition and NO$_x$ emission. This is not surprising given
that atmospheric chemistry is complex and often behaves non-linearly (Fowler et al.,
2007; Fagerli and Aas, 2008). Interactions between the different pollutants,
precipitation variability, changes in the relative amounts and lifetimes of the chemical
species and in gas-particle partitioning all may contribute to the lack of correlation
between emission and deposition trends. Non-linearities between emission and
deposition change have been described also elsewhere (Aguillaume et al., 2016;
Karlsson et al., 2011). Deposition in eastern China is also influenced by emissions
from outside the region, further degrading any expected correlation with local
emissions.
**4.6 Uncertainties and limitations**
The present study examined annual trends of concentrations of N$_r$ species in air
and precipitation as well as dry and bulk N deposition based on Kendall tests and only
five annual data values (2011-2015). Although the test can use as few as 4 data points,
indications of statistically significant trends for datasets are unlikely to be truly
representative of the trends that are actually occurring due to in the short duration of
the measurement dataset. Longer time series (e.g., more than 10-year) will likely
allow detection of more significant time trends in future work. Another uncertainty
may arise from the fact that we used fixed monthly mean dry deposition velocities of
gaseous and particulate N$_r$ species for the same months from June 2013 to December
2015. Nevertheless, the uncertainty in the $V_d$ value did not largely affect the
deposition trend, as the annual trend in dry deposition of N$_r$ species is more likely
driven by changes in ambient N$_r$ concentrations than to changing deposition velocities,
as evident from fairly low standard deviations of annual mean $V_d$ of N$_r$ species at our
selected 27 sites between 2008 and 2012 (~0.029 for NH$_3$, ~0.005 for NO$_2$, ~0.054 for
HNO$_3$, and ~0.019 for both $p$NH$_4^+$ and $p$NO$_3^-$, data were extracted from Zhao et al.

(2017)).





In addition, we did not account for inter-annual changes in meteorology, which
also strongly influences atmospheric $N_r$ levels and N deposition (Xu et al., 2015,
2017). For example, air concentrations of $NO_2$, $NH_3$, and $pNH_4^+$ and $pNO_3^-$ trend to
increase under the relatively stagnant conditions prior to a cold front's arrival and
decrease substantially after the cold front brings precipitation and strong winds into
the region (Xu et al., 2017). On the inter-annual time scale, the frequency of cold front
passages may be affected by large-scale circulation patterns such as the position of the
Siberian high for eastern China (Jia et al., 2015). Given that *in-situ* measurements of
meteorological variables are not available, and that GEOS-5 assimilated
meteorological fields were updated after May 2013, an evaluation of the effect of
meteorology on $N_r$ concentration and deposition is recommended for future work.
Uncertainties also exist in the source attribution calculated with the GEOS-Chem
simulations, since results largely depend on the emission inventories fed to the model.
Zhao et al. (2017) pointed out that uncertainties in current $NH_3$ emissions inventories
(e.g. large range of the emission value in current studies and absence of inclusion of
bi-directional $NH_3$ exchange between the land and atmosphere) may influence
nitrogen deposition simulation in China. Future work based on improved $NH_3$
emission inventories (e.g., Zhang et al., 2018) and including bidirectional ammonia
exchange with the surface is essential to better examine source attribution of N
deposition in China.
**5. Conclusion**
We have characterized spatial and temporal (annual and seasonal) variations in
concentrations and deposition of major $N_r$ species in air ($NH_3$, $NO_2$, $HNO_3$, $pNH_4^+$,
and $pNO_3^-$) and precipitation ($NH_4^+$-N and $NO_3^-$-N) for three land use types (e.g.,
urban, rural and background) in eastern China by examining five-year (2011-2015) *in*
*situ* measurements at twenty-seven sites. We further examined regional features of $N_r$
pollution by comparison of satellite and surface measurements of $NH_3$ and $NO_2$ and
examined the sources of total N deposition over the whole region for the year 2010
using the GEOS-Chem model at horizontal resolution of $1/2^o \times 2/3^o$. Our major
results and conclusions are as follows:





In eastern China, annual mean concentrations and dry and bulk deposition fluxes
of measured $N_r$ species in air and precipitation generally ranked in the order urban >
rural > background. The air concentrations and dry deposition were usually higher at
all land use types in the northern region of eastern China than in the southern region,
especially (except $HNO_3$) at rural sites, for which the differences reached statistically
significant levels. This is also true for the annual VWM concentrations of $NH_4^+$-N,
$NO_3^-$-N, and TIN in precipitation, whereas bulk deposition fluxes of these species
were comparable for matched land use types between the northern and southern
regions.

No significant trends in the annual mean concentrations and dry and bulk

deposition fluxes of measured $N_r$ species in air and precipitation were observed at
almost all sites during the 2011-2015 period. Also, annual averages of these values
showed non-significant changes between the 2011-2012 and 2013-2015 periods for all
land use types. Ambient total concentrations of measured $N_r$ species showed a
non-significant seasonal variation at all land use types, whereas individual $N_r$ species
exhibited a significant seasonal variation in most cases, except for $NO_2$ and $pNH_4^+$ at
urban sites, and $HNO_3$ at all land use types. Unlike air concentrations, dry deposition
of total $N_r$ showed a consistent and significant seasonal variation for each land use
type, with the highest values in summer and the lowest values in winter. The $V_d$ was a
dominant factor influencing seasonal variations of $NO_2$, $HNO_3$, and $pNH_4^+$
concentrations, while seasonal variations of $NH_3$ and $pNO_3^-$ are mainly influenced by
their respective air concentrations. The concentrations of $NH_4^+$-N, $NO_3^-$-N, and TIN in
precipitation showed significant seasonal variations, ranking in a consistent order of
winter > spring > autumn ~ summer. Also, significant seasonal variations in bulk
deposition were also found, following in a consistent order of summer > spring ~
autumn > winter.

Both IASI satellite-retrieved $NH_3$ columns and OMI satellite-retrieved $NO_2$

columns over eastern China showed higher values in the north than in the south. In
addition, significant positive correlations were found between measured $NH_3$
concentrations    and    retrieved    $NH_3$    columns,    and    between    measured    $NO_2$



concentrations and columns. These results together reveal that atmospheric $N_r$ pollution is more serious in the northern region, and also suggest that satellite retrievals of $NH_3$ and $NO_2$ columns can provide useful information on spatial concentration variability of these two key $N_r$ species at a regional or national scale. Weak correlations between IASI_$NH_3$ observations and surface $NH_3$ measurements were found at most selected sites, suggesting that IASI_$NH_3$ observations in their current state are not as readily used to accurately track temporal variability in surface $NH_3$ concentrations.

Ammonia is currently not included in China's emission control policies of air pollution precursors, although the necessity of mitigation has been the subject of discussion during recent years. Across all urban and rural sites, the slopes of the regression relation between $pNH_4^+$ and the sum of $pSO_4^{2-}$ and $pNO_3^-$ were both smaller than unity, indicating control of $NH_3$ emission not only can directly reduce ambient $NH_3$ concentrations, but also lower the formation of $pNH_4^+$ and $pNO_3^-$. Fertilizer use contributed 36% of the total N deposition over eastern China, suggesting reducing $NH_3$ emissions from fertilizer application would be an effective strategy for reducing N deposition. Overall, our findings reveal persistent serious $N_r$ pollution during the 12th FYP period despite implementation of current emission controls, and highlight the importance of $NH_3$ emission control on mitigating future atmospheric $N_r$ concentrations and deposition in eastern China.

**Acknowledgments**

This study was supported by the National Key R&D Program of China (2017YFC0210101 & 2017YFC0210106, 2014BC954202), the National Natural Science Foundation of China (41705130, 41425007, 31421092) as well as the National Ten-thousand Talents Program of China (X.J. Liu).





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
