# Peer review of "and deposition in eastern China"

_Atmospheric Chemistry and Physics, 2018_

## Referee Comment (RC1) · Anonymous Referee #1 · 13 Jun 2018

This paper presented spatial and temporal trends of reactive nitrogen species in air, precipitation and deposition in eastern China. Some of the spatial patterns described in the paper are interesting, such as the higher rural concentrations observed in the northern region compared to the southern region. The paper discusses the need for ammonia emissions policies to reduce reactive nitrogen in air and in deposition. The nitrogen datasets from this ground-based measurement network is valuable; however, a longer dataset needs to be collected before it is suitable for analyzing temporal trends. With only five years of data, this could be the main reason why most of the annual trends were not significant. Another concern that I have is a lack of explanation on the causes of the spatial and temporal trends, which requires analyzing the reactive nitrogen data with other datasets. The discussions seems biased towards ammonia

emissions reductions as a more effective means of reducing reactive nitrogen than NOx and SO2 emissions reductions, but I don't think there is enough evidence in this study supporting this conclusion.

Specific comments

Line 77: Define Nr since this is the first time that it is mentioned in the paper.

Line 83: Be more careful about linking deposition of N to increased greenhouse gas emissions. The referenced article only suggests that the nitrogen cycle is coupled with the carbon cycle and climate variation; however, the latter could be influenced by many factors.

Lines 110-111: The analysis presented by Xu et al. (2015) is quite similar to this study in terms of the measurement network, nitrogen species, time period, and site categories analyzed. The authors should discuss the previous study and explain how this study is different to avoid presenting a duplicate analysis.

Lines 148-156: This is where it might be appropriate to discuss the previous study, Xu et al. (2015), and emphasize the new work that will be shown in this study.

Line 170: Suggest using "and" instead of "resulting in" because this sentence suggests there is a relationship between economic development and nitrogen emissions. If there is such relationship, please elaborate.

Lines 220-221: You need to be clearer about what type of deposition the open sampler collects. Why is it only "some" dry deposition? Isn't the sampler open to the atmosphere which means it is collecting total deposition?

Line 271: The dates here should be January 2011 to 30 September 2014 because you stated in the next sentence that the data after 30 September 2014 were not used.

Lines 347-349: The concentration ranges are not clear. Is it the range of the mean concentration between sites or between years?

[Figure]

Lines 350-352: What is the reason for the lower concentrations at urban sites in the northern region?

Lines 359-365: I suggest analyzing which nitrogen specie was particularly higher between urban and rural sites and between northern and southern regions because this would provide some insight whether the patterns are related to a specific type of emission source.

Lines 371-374: What is the reason for the higher precipitation concentrations in northern rural sites compared to southern rural sites? Is this related to the higher air concentrations of Nr species in northern rural sites?

Lines 383-401: Presenting only the annual trends in the Nr concentrations is not enough. I think that additional analysis with other variables is necessary to attempt to explain the trends in Nr concentrations (e.g. emissions data). As stated in the introduction, one of the goals of this study is to assess the effectiveness of emissions control measures.

Lines 411-416: Any relationships between precipitation concentration and air concentration trends?

Lines 422-436: What is the reason for the seasonal trends? E.g. changes in emissions, meteorology, and/or air mass patterns? I think these other factors need to be analyzed in order to understand what is influencing the seasonal trends.

Line 478: Instead of presenting bulk deposition, is it possible to estimate wet deposition fluxes by subtracting the dry deposition fluxes from bulk deposition? This allows a comparison between wet and dry deposition.

Lines 462-481: How do these deposition fluxes compare to other parts of the world over this recent time period? I also recommend plotting the spatial distribution of the deposition fluxes on a map because it is difficult to get a sense of the spatial patterns from the text and numbers in this paragraph.

Line 572: If you sum dry and wet/bulk deposition fluxes, the total deposition will be overestimated because the bulk deposition already includes dry deposition.

Figure 8: Could you discuss the results in Fig. 8b? All of the previous trends were urban > rural > background. I find it interesting that the trend for the ratio of reduced to oxidized N is reversed. Also, why is this ratio important?

Section 4.1 and Figure 9: The correlation results show there is good agreement between satellite and ground-based observations. Can you quantify the differences using metrics? E.g., what are the percent differences for each month and annually? The correlation may be good, but the actual concentrations can still be different. Given the good relationship between satellite and surface measurements, are long term satellite data available for conducting temporal trend analysis?

Section 4.2: There is too much speculation on the causes of the seasonal trends. Most of the discussion is based on what previous literature reported. I think you need to analyze other datasets to examine the factors affecting the Nr trends.

Line 725: Could you provide the actual emissions amount from x tonnes in 2010 to y tonnes in 2014? Even though the emissions declined by a certain percentage, the actual emissions amount in 2014 might still be very large. If this is the case, then you will likely not observe a significant decrease in Nr concentrations.

Lines 733-734: How much ammonia is emitted relative to NOx and SO2? I would think NOx and SO2 emissions are higher than those of ammonia. If this is the case, wouldn't NOx and SO2 emissions reductions have larger effects on Nr?

Lines 757-773: I don't think you can really say that ammonia emissions reductions are more important than NOx and SO2 emissions reductions. If ammonia emissions have been increasing, why is the Nr concentration in air and precipitation not increasing (many of the trends were not significant in sect. 3.2)? Also, is it possible that the NOx and SO2 emissions reductions are not large enough? See earlier comment about

the actual emissions amount for NOx and SO2 could be very large despite 9-13% decrease in emissions. Is it appropriate to make this conclusion given that five years of data were analyzed? You also discussed how ammonia neutralizes acidic gases and plays a role in limiting Nr. However, it does not mean that this process is more effective than reducing NOx and SO2 emissions which decrease the formation of acidic gases in the first place.

Lines 775-783: This paragraph needs to mention the NOx and SO2 emissions in the northern region especially given the increased emissions for winter heating? How does they compare with ammonia emissions over an annual basis? A map of the spatial distribution of the ammonia emissions and agriculture activity levels would easily demonstrate that these are higher in the northern region.

Line 801: This should be Fig. S12

Line 803: This should be Sect. S2

Lines 799-811: I think the model simulation and results require further analysis and discussion. The model apportions the contributions of various sources to ammonium and nitrate deposition and suggests agricultural activity is the main contributor. There needs to be more details on the model scenario (e.g. NH3 and NOx emissions estimated from the various sources). Is the larger contribution from agriculture due to larger emissions relative to other sources or is it because area sources have larger impact than point sources in the model? Also, to support the idea that NH3 emissions reductions are important in reducing Nr deposition, you could perform a sensitivity analysis using different scenarios of NH3 emissions reductions for future years.

Line 809: What do you mean by improper fertilizer application? Do you mean too excessive? How much fertilizer is applied annually and is this amount much higher than normal? More background on this issue would be useful.

Line 884: Do you have annual precipitation amounts from weather stations, which can

show whether interannual variability in precipitation amounts affect wet deposition?

---

## Referee Comment (RC2) · Anonymous Referee #2 · 13 Jun 2018

This paper presents a statistical summary and discussion of measurements of components of reactive nitrogen (Nr) in the air and in bulk deposition from the 27 sites of a national network that are located in the eastern part of China. The measurement dataset spans the 5-year period from 2011-2015 inclusive. Measurements are also converted into estimates of wet and dry deposition. The authors analyse various spatiotemporal aspects of the concentrations and deposition dataset including seasonality, trends over the 5-year period, and a comparison between sites in the northern half and the southern half of eastern China. The authors supplement the analysis of measurement data with some GEOS-Chem model runs to explore source contributions to Nr in this region. Discussion includes implication for policymakers concerning the different trends in emissions of Nr versus concentrations and deposition of Nr and of the need

to include emissions of NH3 in emissions reductions planning.

The dataset is comprehensive. The presentation of the results is thorough and the text and figures and tables are very clearly presented. There is an extensive discussion. The data are of importance for understanding Nr in eastern China.

Specific comments:

Five years is not a long time period to attempt to discern 'true' long-term trends in concentrations of atmospheric species. The authors recognise that their time period is short in respect of this aspect of their analysis but they could phrase relevant parts of their text to be more cautious about conclusions on long-term trends.

L124: Replace "subsequence" with "subsequent"

L207: It is not clear what is meant by the phrase "where field sampling was carried out after the year 2010". Is this intended to mean that at some sites the measurements did not begin until after 2010?

L271-2: There is a contradiction between a sentence that states that IASI data up until 31 December 2015 was used and the following sentence that states that data only up until 30 September 2014 was used.

Table 1: (1) State in the caption or footnote what the significance test is testing, i.e. that it is testing for significant difference in mean concentration of a pollutant at a given site type between the northern region and the southern region. (2) The footnote should read LUY not LSY to be consistent with column heading.

Figure 2: The reader is referred to Table S1 in the supplement for the number of sites for each land use type in each region, but cannot the reader be directed more easily to Table 1 in the main paper for these numbers?

Figure 3: (1) I assume the data shown are the means for the 5-year period, in which case it may be helpful to make this explicit in the opening sentence thus: "Seasonal

mean concentrations averaged over 2011-2015 of. . ..". (2) As for Figure 2, can the text "in Table S1 in the supplement" be replaced more directly with "in Table 1". (3) The last part of the caption should refer to significant differences between "seasons" not "sites".

Figure 4: The same 3 comments as made above in connection with Figure 3.

Table 2: Same comments as for Table 1.

Figure 5: Can the reader be directed to Table 2, rather than to Table S1 in the supplement, for the number of sites of each type in each region.

Figure 7: Same comments as for Figure 3 (but with substitution of reference to Table 2 rather than to Table 1).

Figure 8: Same comments as for Figure 7.

L598: Rephrase start of sentence to "Eastern China is a highly industrialized. . ."

L 761: In comparing ion balance, presumably the (molar) concentration of NH4+ was compared against the sum of the molar concentrations of NO3- and TWICE the molar concentration of SO42-? The factor 2 is missing from the text and from the axis title of Figure 10f.

---

## Author Comment (AC1) · 10 Jul 2018

This paper presented spatial and temporal trends of reactive nitrogen species in air, precipitation and deposition in eastern China. Some of the spatial patterns described in the paper are interesting, such as the higher rural concentrations observed in the northern region compared to the southern region. The paper discusses the need for ammonia emissions policies to reduce reactive nitrogen in air and in deposition. The nitrogen datasets from this ground-based measurement network is valuable; however, a longer dataset needs to be collected before it is suitable for analyzing temporal trends. With only five years of data, this could be the main reason why most of the annual trends were not significant. Another concern that I have is a lack of explanation on the causes of the spatial and temporal trends, which requires analyzing the reactive nitrogen data with other datasets. The discussions seems biased towards ammonia emissions reductions as a more effective means of reducing reactive nitrogen than $NO_x$ and $SO_2$ emissions reductions, but I don't think there is enough evidence in this study supporting this conclusion.

**Response:** Thanks for the referee's thoughtful and critical comments on our manuscript. Below we provide a point-by-point response to the reviewer' comments and how we have addressed them in the revised manuscript (in blue).

**Specific comments**

Line 77: Define Nr since this is the first time that it is mentioned in the paper.

**Response:** $N_r$ has been defined as "reactive nitrogen" occurring in the first time in the text.

Line 83: Be more careful about linking deposition of N to increased greenhouse gas emissions. The referenced article only suggests that the nitrogen cycle is coupled with the carbon cycle and climate variation; however, the latter could be influenced by many factors.

**Response:** We have deleted "increased greenhouse gas emissions" and the referenced article in the revision.

Lines 110-111: The analysis presented by Xu et al. (2015) is quite similar to this study in terms of the measurement network, nitrogen species, time period, and site

categories analyzed. The authors should discuss the previous study and explain how this study is different to avoid presenting a duplicate analysis.

**Response:** Thank you for this valuable suggestion. In the revised paper, we have added some sentences to discuss the study of Xu et al., 2015), and explain why the current study is different from the previous one. For details, please see our response to next comment (Lines 148-156).

Lines 148-156: This is where it might be appropriate to discuss the previous study, Xu et al. (2015), and emphasize the new work that will be shown in this study.

**Response**: The main purpose of this study was to reveal spatial-temporal (annual and seasonal) patterns of $N_r$ concentrations and deposition based on a full 5-year (2011-2015) measurement at 27 NNDMN sites in eastern China and its northern and southern parts. It also should be noted that, although the study of Xu et al. (2015) and this study both examined the spatial patterns, the regions divided are different. In contrast, the study of Xu et al., 2015 mainly focused on spatial pattern of N deposition at six regions in China, and did not consider seasonal and annual trends. We have added the following sentences in the revision.

**"**Our previous work (Xu et al., 2015) used multiyear measurements (mainly from Jan. 2010 to Sep. 2014) at the 43 sites in the NNDMN, aiming to provide the first quantitative information on atmospheric $N_r$ concentrations and pollution status across China, and to analyze overall fluxes and spatial variations of N deposition in relation to anthropogenic $N_r$ emissions from six regions**".**

Reference:

Xu, W., Luo, X.S., Pan, Y.P., Zhang, L., Tang, A.H., Shen, J.L., Zhang, Y., Li, K.H., Wu, Q.H., Yang, D.W., Zhang, Y.Y., Xue, J., Li, W.Q., Li, Q.Q., Tang, L., Lu, S.H., Liang, T., Tong, Y.A., Liu, P., Zhang, Q., Xiong, Z.Q., Shi, X.J., Wu, L.H., Shi, W.Q., Tian, K., Zhong, X.H., Shi, K., Tang, Q.Y., Zhang, L.J., Huang, J.L., He, C.E., Kuang, F.H., Zhu, B., Liu, H., Jin, X., Xin, Y.J., Shi, X.K., Du, E.Z., Dore, A.J., Tang, S., Collett, J.L., Goulding, K., Sun, Y.X., Ren, J., Zhang, F.S., and Liu, X.J.: Quantifying atmospheric nitrogen deposition through a nationwide monitoring network across China, Atmos. Chem. Phys. 15 (13), 12345–12360, 2015.

Line 170: Suggest using "and" instead of "resulting in" because this sentence suggests there is a relationship between economic development and nitrogen emissions. If there is such relationship, please elaborate.

**Response**: Agree and done.

Lines 220-221: You need to be clearer about what type of deposition the open sampler collects. Why is it only "some" dry deposition? Isn't the sampler open to the atmosphere which means it is collecting total deposition?

**Response**: We ensure that N deposition collected by continuously-open rain gauge refers to wet/bulk deposition, rather than total deposition. Wet/bulk deposition is generally defined as the sum of wet plus some dusts in non-precipitation period (i.e. sedimentary deposition); while dry deposition includes both gases and particles deposition (in which dust or sedimentary deposition is not included). In fact, the wet/bulk plus dry deposition consists of total N deposition without overestimation.

Although N-containing gases and fine particles can be deposited in the 'dry' form to the sampler funnel, the amount of N captured is negligible compared with the dry deposition to plant canopies (Dämmgen et al., 2005; Sutton and Bleeker, 2013). Thus, it is only "some" or small part dry deposition. To make it clearer, "some" was replaced by "incomplete" in the revision.

References:

Dämmgen, U., Erisman, J. W., Cape, J. N., Grünhage, L., and Fowler, D.: Practical considerations for addressing uncertainties in monitoring bulk deposition, Environ. Pollut. 134(3), 535–548, 2004.

Sutton, M.A., and Bleeker, A.: Environmental science: the shape of nitrogen to come. Nature 494, 435–437, 2013.

Line 271: The dates here should be January 2011 to 30 September 2014 because you stated in the next sentence that the data after 30 September 2014 were not used.

**Response**: This was a wrong expression in the sentence. Actually, we used the daily IASI-NH$_3$ data from 1 January 2011 to 31 December 2015 for the spatial analysis, and from January 2011 to 30 September 2014 for temporal analysis.

We now state that "The daily IASI-NH$_3$ data (provided by the Atmospheric

Spectroscopy Group at Université Libre De Bruxelles, data available at http://iasi.aeris-data.fr/NH$_3$/) from 1 January 2011 to 31 December 2015 was used for the spatial analysis in the present study. For the temporal analysis, we used the IASI_NH$_3$ from 1 January 2011 to 30 September 2014 because an update of the input meteorological data on 30 September 2014 had caused a substantial increase in the retrieved atmospheric NH$_3$ columns."

Lines 347-349: The concentration ranges are not clear. Is it the range of the mean concentration between sites or between years?

**Response**: The ranges of mean concentrations denote the minimum and maximum 5-year mean concentrations of measured five N$_r$ species (i.e., NH$_3$, NO$_2$, HNO$_3$, $p$NH$_4^+$, and $p$NO$_3^-$) for each land use type (i.e., urban, rural and background), which can be derived from Table 1. For example, the values of 1.6 ± 0.2 and 10.2 ± 1.0 µg N m$^{-3}$ are 5-year mean concentrations of HNO$_3$ and NO$_2$ at urban sites in eastern China, respectively.

To make it clear, in the revision we now state that "In eastern China, annual mean concentrations of NH$_3$, NO$_2$, HNO$_3$, $p$NH$_4^+$, and $p$NO$_3^-$ at the urban sites (averages for the 5-year, 1.6 ± 0.2 (for HNO$_3$) to 10.2 ± 1.0 (for NO$_2$) µg N m$^{-3}$) increased by 18, 70, 33, 23, and 43%, respectively, compared with their corresponding concentrations at the rural sites (1.2 ± 1.0 (for HNO$_3$) to 7.2 ± 0.9 (for NH$_3$) µg N m$^{-3}$); they also increased by 78-118% compared with the concentrations at the background sites (0.9 ± 0.1 (for HNO$_3$) to 5.2 ± 0.3 (for NO$_2$) µg N m$^{-3}$) (Table 1)."

Lines 350-352: What is the reason for the lower concentrations at urban sites in the northern region?

**Response**: This is mainly due to the fact that the North China Plain (NCP, that is, the plain areas in Beijing, Tianjin, Hebei, Henan, and Shandong provinces) is located in the northern region. The Plain (i.e., NCP) is featured by intensive agricultural production in rural areas, which contributes 30-40% of the total annual NH$_3$ emissions in China (Huang et al., 2012). In addition, the north is dominated by calcareous soils, which favor high soil NH$_3$ volatilization from croplands (Huang et al., 2015). Those emitted NH$_3$ can directly enhance ambient NH$_3$ concentration and also particulate

$NH_4^+$ concentrations via chemical reactions between $NH_3$ and acidic gases in the atmosphere (e.g., $H_2SO_4$ and $HNO_3$).

References:

Huang, X., Song, Y., Li, M. M., Li, J. F., Huo, Q., Cai, X. H., Zhu, T., Hu, M., and Zhang, H. S: A high-resolution ammonia emission inventory in China, Global Biogeochem. Cycles 26, GB1030, 2012.

Huang, P., Zhang, J. B., Xin, X. L., Zhu, A. N., Zhang, C. Z., Ma, D. H., Zhu, Q. G., Yang, S., and Wu, S. J.: Proton accumulation accelerated by heavy chemical nitrogen fertilization and its long-term impact on acidifying rate in a typical arable soil in the Huang-Huai-Hai Plain, J. Integr. Agric. 14, 148–157, 2015.

Lines 359-365: I suggest analyzing which nitrogen species was particularly higher between urban and rural sites and between northern and southern regions because this would provide some insight whether the patterns are related to a specific type of emission source.

**Response**: Good point. In the old version, we have made a comparison of annual mean concentration of each $N_r$ species between urban and rural sites, as shown in Table 1. In Results Section, we also stated that "In eastern China, annual mean concentrations of $NH_3$, $NO_2$, $HNO_3$, $pNH_4^+$, and $pNO_3^-$ at the urban sites ($1.6 \pm 0.2$ to $10.2 \pm 1.0$ μg N m$^{-3}$) were 18-70% and 78-118% higher than their corresponding concentrations at the rural ($1.2 \pm 1.0$ to $7.2 \pm 0.9$ μg N m$^{-3}$) and background ($0.9 \pm 0.1$ to $5.2 \pm 0.3$ μg N m$^{-3}$) sites, respectively.". According to suggestion by the reviewer, the sentence was revised to make it clearer, and now reads as "In eastern China, annual mean concentrations of $NH_3$, $NO_2$, $HNO_3$, $pNH_4^+$, and $pNO_3^-$ at the urban sites (averages for the 5-year, $1.6 \pm 0.2$ (for $HNO_3$) to $10.2 \pm 1.0$ (for $NO_2$) μg N m$^{-3}$) increased by 18, 70, 33, 23, and 43%, respectively, compared with their corresponding concentrations at the rural sites ($1.2 \pm 1.0$ (for $HNO_3$) to $7.2 \pm 0.9$ (for $NH_3$) μg N m$^{-3}$); they also increased by 78-118% compared with the concentrations at the background sites ($0.9 \pm 0.1$ (for $HNO_3$) to $5.2 \pm 0.3$ (for $NO_2$) μg N m$^{-3}$) (Table 1)."

As for comparisons between northern and southern regions, we added the following sentence in the revision.

"Averaged across three land use types, the annual mean $N_r$ concentrations of five $N_r$ species in the north increased to varying extent (by 84% for $pNO_3^-$, 63% for $pNH_4^+$, 57% for $NH_3$, 47% for $NO_2$, and 28% for $HNO_3$) compared with those in the south.".

Lines 371-374: What is the reason for the higher precipitation concentrations in northern rural sites compared to southern rural sites? Is this related to the higher air concentrations of Nr species in northern rural sites?

**Response**: Yes, it is mainly due to significantly ($p<0.05$) higher air concentrations of five $N_r$ species at northern rural sites than at southern rural sites (Table 1), as $NH_4^+$-N and $NO_3^-$-N in precipitation primarily originates from reduced N (e.g., gaseous $NH_3$ and particulate $NH_4^+$) and oxidized N (e.g., gaseous $NO_2$, $HNO_3$, and particulate $NO_3^-$) in air (Wang et al., 2018). Another reason is the "concentration effect" because annual precipitation is much lower in the north (e.g. 400-600 mm per year) than in the south (e.g. 800-1400 mm per year).

Reference:

Wang, H.B., Shi, G.M., Tian, M., Chen, Y.., Qiao, B.Q., Zhang, L.Y., Yang, F.M., Zhang, L.M., and Luo, Q.: Wet deposition and sources of inorganic nitrogen in the Three Gorges Reservoir Region, China, Environ. Pollut., 233, 520-528, 2018.

Lines 383-401: Presenting only the annual trends in the Nr concentrations is not enough. I think that additional analysis with other variables is necessary to attempt to explain the trends in Nr concentrations (e.g. emissions data). As stated in the introduction, one of the goals of this study is to assess the effectiveness of emissions control measures.

**Response:** We partly agree with the referee. Given that $N_r$ ($NH_3$ and $NO_2$) emissions and concentrations are in different units, and higher $N_r$ concentrations generally result in higher N deposition on an annual timescale, the comparison of $N_r$ emissions with deposition (both are calculated in the unit of kg N $ha^{-1}$ $yr^{-1}$) is more reasonable relative to the comparison between $N_r$ emissions and concentrations. As the main objective of this study is to spatial-temporal patterns of atmospheric inorganic N concentrations and deposition, we presented relevant results of $N_r$ concentrations and deposition in the Results, and put the comparison between $N_r$ emission and deposition

in the Discussions (please see Section 4.5 in the old version). Therefore, we keep the analysis as it is.

According to the referee's suggestion, here we also attempt to make the corresponding comparisons using the annual average values on $N_r$ emissions ($NH_3$ and $NO_x$) and air concentrations of $NH_3$ and $NO_2$ at the sixteen sites (details are given in Section 4.5). As shown in Figure 1 below, across all the sites annual mean $NH_3$ emissions and concentrations showed increases of 4 and 20% in 2013-2015 compared with those in 2011-2015, respectively. Correspondingly, annual mean $NO_x$ emissions and $NO_2$ concentrations showed reductions of 18 and 2%, respectively. In addition, there were no significant ($p>0.05$) correlations between $NH_3$ emissions and concentrations, and between $NO_x$ emissions and concentrations during 2011-2015.

[Figure]

Figure 1. Comparisons of $NH_3$ emissions and $NH_3$ concentrations, and $NO_x$ emissions and $NO_2$ concentrations between the periods 2011-2012 and 2013-2015.

Lines 411-416: Any relationships between precipitation concentration and air concentration trends?

**Response:** Based on analysis of annual averages at the sixteen sites with continuous and simultaneous measurements of dry and wet/bulk N deposition during 2011-2015 (site names are given in Fig. S6 and Table S1), a positive relationship (r=0.62, $p$=0.27) was found between $NH_4^+$-N concentrations in precipitation and air concentrations of reduced $N_r$ (the sum of $NH_3$ and particulate $NH_4^+$), whereas a negative relationship (r= -0.85, $p$=0.07) was found between $NO_3^-$-N concentration in precipitation and air concentration of oxidized $N_r$ (the sum of $NO_2$, $HNO_3$, and particulate $NO_3^-$). We think

that those findings are acceptable. This is because that $NH_3$ is locally deposited and relatively high $NH_3$ concentration generally distributed near emission sources. In contrast, local oxidized $N_r$ concentration can be affected by atmospheric transport from nearby regions. No significant correlations between precipitation concentration and air concentration are mainly due to relatively small changes in $NH_3$ and $NO_x$ emissions (Fig. 12) and annual mean precipitation amount (from 800 to 951 mm, Fig. S14) during 2011-2015.

Lines 422-436: What is the reason for the seasonal trends? E.g. changes in emissions, meteorology, and/or air mass patterns? I think these other factors need to be analyzed in order to understand what is influencing the seasonal trends.

**Response:** Thank you for pointing it out, and we have analyzed the seasonal trends of $N_r$ concentrations integrated with changes air mass trajectory (please see added context in Section 4.2). As for $N_r$ emissions, it is well known that $NH_3$ emissions in China typically peaked in summer due to the summertime application of fertilizer for double cropping in together with higher temperature, and the lowest values occurred in winter (Paulot et al., 2014; Kang et al., 2016; Zhang et al., 2018). In contrast, the highest $NO_2$ emissions generally occur in winter because of domestic heating needs, and minimum values generally occur in spring (Zhang et al., 2007). Thus, we directly used previous literature reported to explain corresponding results in the present study.

References:

Paulot, F., Jacob, D.J., Pinder, R.W., Bash, J.O., Travis, K., and Henze, D.K.: Ammonia emissions in the United States, European Union, and China derived by high-resolution inversion of ammonium wet deposition data: Interpretation with a new agricultural emissions inventory (MASAGE_NH$_3$), J. Geophys. Res. Atmos., 119, 4343–4364, https://doi:10.1002/2013JD021130, 2014.

Kang, Y. N., Liu, M. X., Song, Y., Huang, X., Yao, H., Cai, X. H., Zhang, H. S., Kang, L., Liu, X. J., Yan, X. Y., He, H., Zhang, Q., Shao, M., and Zhu, T.: High-resolution ammonia emissions inventories in China from 1980 to 2012, Atmos. Chem. Phys., 16, 2043–2058, 2016.

Zhang, Q., Streets, D. G., He, K., Wang, Y., Richter, A., Burrows, J. P., Uno, I., Jang,

C. J., Chen, D., Yao, Z., and   Lei, Y.: NO$_x$ emission trends for China, 1995-2004: The view from the ground and the view from space, J. Geophys. Res., 112, D22306, 2007.

Zhang, L., Chen, Y. F., Zhao, Y. H., Henze, D. K., Zhu, L. Y., Song, Y., Paulot, F., Liu, X. J., Pan, Y. P., and Huang, B. X.: Agricultural ammonia emissions in China: reconciling bottom-up and top-down estimates, Atmos. Chem. Phys., 18, 339–355, 2018.

Line 478: Instead of presenting bulk deposition, is it possible to estimate wet deposition fluxes by subtracting the dry deposition fluxes from bulk deposition? This allows a comparison between wet and dry deposition.

**Response:** Our previous work (Liu et al., 2006; Zhang et al., 2008) showed the ratios of wet-only and bulk deposition of inorganic N being 0.68-0.93 in North China Plain. Therefore it seems not possible to estimate wet deposition fluxes by multiplying a coefficient or subtracting the dry deposition fluxes from bulk deposition, since fraction of dry deposited N in bulk deposition is variable and not fixed across monitoring years. Anyway, we mentioned this in the revision.

References:

Liu X.J., Ju X.T., Zhang Y., He C.E., Kopsch J., and Zhang F.S.: Nitrogen deposition in agroecosystems in the Beijing area. Agriculture, Ecosystems & Environment 113, 370-377, 2006.

Zhang Y., Liu X.J., Fangmeier A., Goulding K.T.W., and Zhang F.S.: Nitrogen inputs and isotopes in precipitation in the North China Plain. Atmospheric Environment 42, 1436-1448, 2008.

Lines 462-481: How do these deposition fluxes compare to other parts of the world over this recent time period? I also recommend plotting the spatial distribution of the deposition fluxes on a map because it is difficult to get a sense of the spatial patterns from the text and numbers in this paragraph.

**Response:** On the basis of 2001 ensemble-mean modeling results from 21 global chemical transport models (Vet et al., 2014), three global N deposition hotspots were western Europe (with levels from 20.0 to 28.1 kg N ha$^{-1}$ yr$^{-1}$, South Asia (Pakistan,

India, and Bangladesh) from 20.0 to 30.6 kg N ha$^{-1}$ yr$^{-1}$ and East Asia from 20 to 38.6 kg N ha$^{-1}$ yr$^{-1}$ in eastern China (the global maximum). Extensive areas of high deposition from 10 to 20 kg N ha$^{-1}$ yr$^{-1}$ appear in the eastern United States and southeastern Canada as well as most of central Europe. Obviously, our estimated total N deposition fluxes (dry plus wet/bulk deposition, averaging from 34.2 kg N ha$^{-1}$ yr$^{-1}$ at background sites to 59.7 kg N ha$^{-1}$ yr$^{-1}$ at urban sites, Table 1) showed a much higher values. Relevant comparisons have been reported in our previous work (Xu et al., 2015).

As for data presentation, we think that the use of Table is reasonable and useful due to following two reasons. First, our analysis was based on land use types rather than single sampling site, and thus it is impractical to plot the spatial distribution of the deposition fluxes on a map. Second, using Tables can directly provide basic data for scientific communities for carrying out other relevant research. Therefore, we keep the Table as it is.

References:

Xu, W., Luo, X.S., Pan, Y.P., Zhang, L., Tang, A.H., Shen, J.L., Zhang, Y., Li, K.H., Wu, Q.H., Yang, D.W., Zhang, Y.Y., Xue, J., Li, W.Q., Li, Q.Q., Tang, L., Lu, S.H., Liang, T., Tong, Y.A., Liu, P., Zhang, Q., Xiong, Z.Q., Shi, X.J., Wu, L.H., Shi, W.Q., Tian, K., Zhong, X.H., Shi, K., Tang, Q.Y., Zhang, L.J., Huang, J.L., He, C.E., Kuang, F.H., Zhu, B., Liu, H., Jin, X., Xin, Y.J., Shi, X.K., Du, E.Z., Dore, A.J., Tang, S., Collett, J.L., Goulding, K., Sun, Y.X., Ren, J., Zhang, F.S., and Liu, X.J.: Quantifying atmospheric nitrogen deposition through a nationwide monitoring network across China. Atmos. Chem. Phys. 15 (13), 12345–12360, 2015.

Vet, R., Artz, R. S., Carou, S., Shaw, M., Ro, C.-U., Aas, W., Baker, A., Bowersox, V. C., Dentener, F., Galy-Lacaux, C., Hou, A., Pienaar, J. J., Gillett, R., Forti, M. C., Gromov, S., Hara, H., Khodzher, T., Mahowald, N. M., Nickovic, S., Rao, P. S. P., and Reid, N. W.: A global assessment of precipitation chemistry and deposition of sulfur, nitrogen, sea salt, base cations, organic acids, acidity and pH, and phosphorus, Atmos. Environ., 93, 3–100, 2014.

Line 572: If you sum dry and wet/bulk deposition fluxes, the total deposition will be

overestimated because the bulk deposition already includes dry deposition.

**Response:** This concern was answered in our previous response to "Lines 220-221". In fact, our wet/bulk (including wet plus sedimentary deposition) + dry deposition (gases plus fine particles (non-sedimentary) deposition) denote a complete total N deposition. This means the wet/bulk deposition is not pure 'wet' deposition while the dry deposition is not complete 'dry' deposition. According to our previous studies (Liu et al., 2006; Zhang et al., 2008), annual difference between bulk and wet deposition was 1.3-9.6 kg N ha$^{-1}$ in northern Chinese agroecosystems. Therefore, to avoid misunderstanding, we defined the total N deposition as the sum of dry and bulk deposition in this study, although it is in principle defined as the sum of dry and wet deposition.

References:

Liu, X.J., Ju, X.T., Zhang, Y., He, C.E., Kopsch, J., and Zhang, F.S.: Nitrogen deposition in agroecosystems in the Beijing area, Agr. Ecosyst. Environ. 113(1), 370–377, doi:10.1016/j.agee.2005.11.002, 2006.

Zhang, Y., Liu, X. J., Fangmeier, A., Goulding, K. T. W., and Zhang, F. S.: Nitrogen inputs and isotopes in precipitation in the North China Plain, Atmos. Environ., 42, 1436–1448, 2008.

Figure 8: Could you discuss the results in Fig. 8b? All of the previous trends were urban > rural > background. I find it interesting that the trend for the ratio of reduced to oxidized N is reversed. Also, why is this ratio important?

**Response:** The opposite trend for the ratio of reduced to oxidized N is reasonable, as it depends on proportion of reduced and oxidized N deposition in the total deposition. This ratio can be used to indicate the relative contribution of $N_r$ from agricultural and industrial activities to N deposition (Xu et al., 2015) because the major anthropogenic source of reduced N ($NH_3$ and particulate $NH_4^+$) is mainly affected by $NH_3$ volatilized from animal excrement and the application of nitrogenous fertilizers in agriculture, while anthropogenic sources of oxidized N ($NO_2$, $HNO_3$ and particulate $NO_3^-$) is primarily dominated by $NO_x$ emitted from fossil fuel combustion in transportation, power plant, and factories.

As shown in Fig. 8b, the averaged ratios at three land use types were slightly higher in the 2013-2015 period than in the 2011-2012 period, indicating agricultural $NH_3$ emission played a more and more important role in N deposition. This result, in turn, supports our conclusion from sensitivity tests by the GEOS-Chem model that mitigation of agricultural $NH_3$ emissions should be a priority to tackle serious N deposition in eastern China.

As suggested by the referee, we added the following discussion in the revision (in the Section 4.4):

"This conclusion to some extent is supported by increased ratios of the ratio of reduced to oxidized N in the total deposition at three land use types (Fig. 8b), as the major anthropogenic source of reduced N is mainly affected by $NH_3$ volatilized from animal excrement and the application of nitrogenous fertilizers in agriculture. Absence of $NH_3$ emission controls may be the main reason for a small and non-significant change in the total N deposition between 2011-12 and 2013-15 (Fig. S6, Supplement), despite enforcement of stringent emission controls on $NO_x$ and $SO_2$."

Reference:

Xu, W., Luo, X.S., Pan, Y.P., Zhang, L., Tang, A.H., Shen, J.L., Zhang, Y., Li, K.H., Wu, Q.H., Yang, D.W., Zhang, Y.Y., Xue, J., Li, W.Q., Li, Q.Q., Tang, L., Lu, S.H., Liang, T., Tong, Y.A., Liu, P., Zhang, Q., Xiong, Z.Q., Shi, X.J., Wu, L.H., Shi, W.Q., Tian, K., Zhong, X.H., Shi, K., Tang, Q.Y., Zhang, L.J., Huang, J.L., He, C.E., Kuang, F.H., Zhu, B., Liu, H., Jin, X., Xin, Y.J., Shi, X.K., Du, E.Z., Dore, A.J., Tang, S., Collett, J.L., Goulding, K., Sun, Y.X., Ren, J., Zhang, F.S., and Liu, X.J.: Quantifying atmospheric nitrogen deposition through a nationwide monitoring network across China, Atmos. Chem. Phys. 15 (13), 12345–12360, 2015.

Section 4.1 and Figure 9: The correlation results show there is good agreement between satellite and ground-based observations. Can you quantify the differences using metrics? E.g., what are the percent differences for each month and annually? The correlation may be good, but the actual concentrations can still be different. Given the good relationship between satellite and surface measurements, are long

term satellite data available for conducting temporal trend analysis?

**Response:** It is difficult to quantify the differences between satellite and ground-based observations using a uniform unit. Since ground and satellite measurements give the mixing ratios of $N_r$ species ($NH_3$ and $NO_2$) in the surface layer and tropospheric integrated column densities of the species, respectively, estimating the satellite-derived ground concentrations of $N_r$ species required their corresponding vertical profiles. Unfortunately, measurements of vertical profiles of concentrations above the surface are rare. On this point, in earlier version we stated in the text "To make a more accurate comparison, the vertical profile is recommended to convert the columns to the ground concentrations in future work". Alternatively, we analyzed the correlations between satellite and ground-based observations to detect whether there is a consistency in spatial and temporal distributions.

As for temporal analysis, the following paragraph in the Section 4.1 can answer whether long term satellite data are available for conducting temporal trend analysis.

"…the OMI_NO$_2$ retrieval can well capture the temporal variations of surface $NO_2$ concentrations over eastern China, whereas the IASI_NH$_3$ retrievals better capture temporal variability in surface concentrations for the northern region. The weak correlations observed between IASI_NH$_3$ observations and surface measurements at ten of the fourteen sites in the southern region (Fig. S7, Supplement) suggest that the IASI_NH$_3$ observations need to be improved for investigating temporal variability in $NH_3$ concentration, despite that the satellite observation is at a specific time of day while the surface concentrations integrate across the diurnal cycle of emissions and mixing layer evolution."

Section 4.2: There is too much speculation on the causes of the seasonal trends. Most of the discussion is based on what previous literature reported. I think you need to analyze other datasets to examine the factors affecting the Nr trends.

**Response:** In the revision, we analyzed datasets of air mass trajectory to examine influence of potential atmospheric transport on the resulting seasonal $N_r$ trends. The following paragraphs were added as follows:

"In order to identify potential transport of $NO_2$, $p$NH$_4^+$ and $p$NO$_3^-$ from northern

region, we calculated three-day backward trajectories arriving at five southern sites (Nanjing, Baiyun, Taojing, Ziyang and Huinong) during January, April, July and October using the TrajStat. The TrajStat analysis generally showed that the high proportions (overall 10-36%) of air masses from the north to the south of eastern China occurred in the autumn/winter, suggesting that the transport of $NO_2$, $pNH_4^+$ and $pNO_3^-$ from northern China would result in increases in their respective concentrations in autumn/winter south of the Qinling Mountains-Huaihe River line, except at Ziyang site (Fig. S14, Supplement).

Line 725: Could you provide the actual emissions amount from x tonnes in 2010 to y tonnes in 2014? Even though the emissions declined by a certain percentage, the actual emissions amount in 2014 might still be very large. If this is the case, then you will likely not observe a significant decrease in Nr concentrations.

**Response:** In the revised paper, we added the actual emission amount for the years 2010 and 2014. We now state that "…total annual emissions of $SO_2$ and $NO_x$ were reduced by 12.9% and 8.6% in 2014 (approximately 9.9 Tg S $yr^{-1}$ and 6.3 Tg N $yr^{-1}$, respectively), respectively, compared with those in 2010 (approximately 11.3 Tg S $yr^{-1}$ and 6.9 Tg N $yr^{-1}$, respectively)".

Yes, since $NO_x$ emissions were still at high level in 2014. We did not find a significant decrease in $NO_2$ concentrations in the current study. For total $N_r$, persistent high concentrations is likely due to the absence of $NH_3$ regulations, as $NH_3$ emission reduction had a larger influence on $N_r$ concentration (for details, please see our response to next comment to Lines 733-734)

Lines 733-734: How much ammonia is emitted relative to NOx and SO2? I would think NOx and SO2 emissions are higher than those of ammonia. If this is the case, wouldn't NOx and SO2 emissions reductions have larger effects on Nr?

**Response:** Yes, total annual emissions of $NO_x$ and $SO_2$ (average over 2011-2015, approximately 7.0 Tg N $yr^{-1}$ and 9.8 Tg S $yr^{-1}$) were higher than those of $NH_3$ emission (10.0 Tg N $yr^{-1}$) during the period of 2011-2015 in eastern China (details of emission data are given in Section 4.5). In addition, the annual molar ratios of

(2SO$_2$+NO$_x$)/NH$_3$ were greater than 1 (ranging from 1.3 to 1.8) during the period. These results suggest that NH$_3$ emissions presented the limiting factor to the formation of secondary inorganic ions (e.g., particulate NH$_4^+$ and NO$_3^-$), and its emission reductions have large effects on N$_r$ (e.g., gaseous NH$_3$ and particulate NH$_4^+$ and NO$_3^-$). This is also true at the national scale, as the molar amount of (2SO$_2$+NO$_x$) still substantially exceeded that of NH$_3$ at least until 2015 (Zhang et al., 2017).

Reference:

Zhang, X. M., Wu, Y. Y., Liu, X. J., Reis, S., Jin, J. X., Dragosits, U., Damme, Van M., Clarisse, L., Whitburn, S., and Coheur, P. F.: Ammonia emissions may be substantially underestimated in China, Environ. Sci. Techno., 51, 12089-12096, 2017.

Lines 757-773: I don't think you can really say that ammonia emissions reductions are more important than NOx and SO2 emissions reductions. If ammonia emissions have been increasing, why is the Nr concentration in air and precipitation not increasing (many of the trends were not significant in sect. 3.2)? Also, is it possible that the NOx and SO2 emissions reductions are not large enough? See earlier comment about the actual emissions amount for NOx and SO2 could be very large despite 9-13% decrease in emissions. Is it appropriate to make this conclusion given that five years of data were analyzed? You also discussed how ammonia neutralizes acidic gases and plays a role in limiting Nr. However, it does not mean that this process is more effective than reducing NOx and SO2 emissions which decrease the formation of acidic gases in the first place.

**Response:** Based on the discussions in Lines 757-773, we did not give the viewpoint that NH$_3$ emissions reductions are more important than NO$_x$ and SO$_2$ emissions reductions. We concluded that implementation of NH$_3$ control strategies, relative to current NO$_x$ and SO$_2$ emission controls, should be considered to mitigate atmospheric N$_r$ pollution. Between the periods 2013-2015 and 2011-2012, the mean concentrations of NH$_3$ and $p$NH$_4^+$ overall showed non-significant increases (10-38%) at all land use types, whereas small changes in remaining N$_r$ species occurred. As a result, annual total N$_r$ concentration in air showed increases to varying extent at three land use types.

This also highlights the importance of $NH_3$ emission reduction in controlling $N_r$ pollution. Indeed, for individual species small changes in air concentrations of $NO_2$, $HNO_3$ and $pNO_3^-$ may be due to that the $NO_x$ and $SO_2$ emissions reductions are not large enough.

To avoid misunderstanding, we now state that "implementation of $NH_3$ control strategies, together with more stringent $NO_x$ and $SO_2$ emission controls, should be considered to mitigate atmospheric $N_r$ pollution."

Lines 775-783: This paragraph needs to mention the NOx and SO2 emissions in the northern region especially given the increased emissions for winter heating? How does they compare with ammonia emissions over an annual basis? A map of the spatial distribution of the ammonia emissions and agriculture activity levels would easily demonstrate that these are higher in the northern region.

**Response:** Thank you for this suggestion. We added the following discussions in Section 4.4 in the revision.

"In addition, higher $NH_3$ concentration is also likely due to the higher $NH_3$ volatilization in calcareous soils than that in the acidic red soil, as mentioned in Section 2.1. Total annual $NH_3$ emissions in northern region increased from 4.3 Tg N $yr^{-1}$ in 2011 to 4.7 Tg N $yr^{-1}$ at an annual rate of 1.8%. In contrast, the emissions of $NO_x$ and $SO_2$ averaged 2.8 Tg N $yr^{-1}$ and 3.7 Tg S $yr^{-1}$ during 2011-2015, and decreased at annual rates of 6.8 and 5.7%, respectively (details of the emissions will be illustrated in Section 4.5). Such reductions may enhance free $NH_3$ in the atmosphere. However, according to a modeling study by Han et al. (2017), the influence of removing anthropogenic $SO_2$ emissions on dry N deposition fluxes during 2010-2014 was quite weak, with the change within -0.5~0.5 (kg N $ha^{-1}$ $yr^{-1}$) over most regions in China."

We think that current discussion is sufficient to explain why total dry N deposition fluxes at three land use types were higher in the northern region of eastern China than in the southern region. Given that the article is already relatively lengthy and this part of discussion is not the core, we did not compare the spatial distribution of the ammonia emissions and agriculture activity levels in eastern China in the revision.

Reference:

Han, X., Zhang, M. G., Skorokhod, A., and Kou, X. X.: Modeling dry deposition of reactive nitrogen in China with RAMS-CMAQ, Atmos. Environ., 166, 47–61, 2017.

Line 801: This should be Fig. S12

**Response**: Corrected.

Line 803: This should be Sect. S2

**Response**: Corrected.

Lines 799-811: I think the model simulation and results require further analysis and discussion. The model apportions the contributions of various sources to ammonium and nitrate deposition and suggests agricultural activity is the main contributor. There needs to be more details on the model scenario (e.g. NH3 and NOx emissions estimated from the various sources). Is the larger contribution from agriculture due to larger emissions relative to other sources or is it because area sources have larger impact than point sources in the model? Also, to support the idea that NH3 emissions reductions are important in reducing Nr deposition, you could perform a sensitivity analysis using different scenarios of NH3 emissions reductions for future years.

**Response:** Thank you for this suggestion. The larger contribution from agriculture is due to larger emissions relative to other sources. In the revised paper, we now state that "The total $NH_3$ and $NO_x$ emissions from each source over eastern China and its contribution to total emissions in China are presented in Table S13 in the Supplement. The $NH_3$ and $NO_x$ emissions over eastern China are 11.6 Tg N yr$^{-1}$ and 8.5 Tg N yr$^{-1}$ in 2010, which, respectively, account for 90% and 89% of their total emissions over China. Agricultural sources, including fertilizer use and livestock, comprise most of the $NH_3$ emissions while fuel combustion activities, including industry, power plant, and transportation contribute most of the $NO_x$ emissions and small amounts of $NH_3$ emissions. Both $NH_3$ and $NO_x$ have natural sources (including lightning, biomass burning and soil emissions), but are negligible compared to anthropogenic emissions over eastern China."

Based on outputs from the model simulation, it is obvious that controlling agricultural

NH$_3$ emission can undoubtedly lower N deposition. Thanks for the suggestion on performance of scenarios analysis of NH$_3$ emission reduction, we conducted a separate model simulation which reduce emissions from fertilizer use by 20%.We add the following sentences in the text:

"To test the importance of future ammonia emission control strategies, we conducted separate model simulations which reduced NH$_3$ emissions from fertilizer use by 20%. The results showed that a 20% reduction in fertilizer NH$_3$ emissions can lead to a 7.4% decrease in total N deposition over Eastern China"

In future study, we will attempt to use improved NH$_3$ emission (e.g., Zhang et al., 2018) inventories to detail the relative contribution of emissions sources to N deposition and further scenarios analysis of NH$_3$ emissions.

Reference:

Zhang, L., Chen, Y. F., Zhao, Y. H., Henze, D. K., Zhu, L. Y., Song, Y., Paulot, F., Liu, X. J., Pan, Y. P., and Huang, B. X.: Agricultural ammonia emissions in China: reconciling bottom-up and top-down estimates, Atmos. Chem. Phys., 18, 339–355, 2018.

Line 809: What do you mean by improper fertilizer application? Do you mean too excessive? How much fertilizer is applied annually and is this amount much higher than normal? More background on this issue would be useful.

**Response:** "improper fertilizer application" means N fertilizers were not applied in appropriate fertilization pattern (e.g., fertilizing with a suitable choice of chemical, at the correct application level, selecting the best of the year and location). To make it clear, we now state that "These results indicate that reducing NH$_3$ emissions by use of appropriate fertilization patterns (e.g., 4 R technologies (Right amount, Right time, Right form and Right application technique), Ju et al., 2009) should be a priority in curbing N deposition in eastern China".

Reference:

Ju, X.T., Xing, G.X., Chen, X.P., Zhang, S.L., Zhang, L.J., Liu, X.J., Cui, Z.L., Yin, B., Christie, P., Zhu, Z.L., and Zhang, F.S.: Reducing environmental risk by improving N management in intensive Chinese agricultural systems, Proc. Natl.

Acad. Sci. U. S. A. 106, 3041-3046, 2009.

Line 884: Do you have annual precipitation amounts from weather stations, which can show whether interannual variability in precipitation amounts affect wet deposition?

**Response:** We measured precipitation amounts at 27 study sites during 2011-2015. According to suggestion by the referee, we selected 16 sites with continuous 5-year measurements, and our results demonstrated an obvious interannual variability in precipitation amounts. Thus, wet deposition to some extent can be affected by the change in precipitation amounts.

In the revised paper, we added Figure S14 in the Supplement, and stated in the text that "For example, a large inter-annual variation in precipitation amount was observed at the selected 16 sites during 2011-2015, which partially lead to inter-annual changes in wet/bulk N deposition.

---

## Author Comment (AC2) · 10 Jul 2018

This paper presents a statistical summary and discussion of measurements of components of reactive nitrogen (Nr) in the air and in bulk deposition from the 27 sites of a national network that are located in the eastern part of China. The measurement dataset spans the 5-year period from 2011-2015 inclusive. Measurements are also converted into estimates of wet and dry deposition. The authors analyse various spatiotemporal aspects of the concentrations and deposition dataset including seasonality, trends over the 5-year period, and a comparison between sites in the northern half and the southern half of eastern China. The authors supplement the analysis of measurement data with some GEOS-Chem model runs to explore source contributions to Nr in this region. Discussion includes implication for policymakers concerning the different trends in emissions of Nr versus concentrations and deposition of Nr and of the need to include emissions of NH3 in emissions reductions planning. The dataset is comprehensive. The presentation of the results is thorough and the text and figures and tables are very clearly presented. There is an extensive discussion. The data are of importance for understanding Nr in eastern China.

**Response**: Thanks for the recognition of our contribution. Below we provide a point-by-point response to the species comments, together with proposed changed in the revised manuscript (in blue).

Specific comments:

Five years is not a long time period to attempt to discern 'true' long-term trends in concentrations of atmospheric species. The authors recognise that their time period is short in respect of this aspect of their analysis but they could phrase relevant parts of their text to be more cautious about conclusions on long-term trends.

**Response:** The suggestion has been implemented in the revision.

L124: Replace "subsequence" with "subsequent"

**Response:** Agree and done.

L207: It is not clear what is meant by the phrase "where field sampling was carried out after the year 2010". Is this intended to mean that at some sites the measurements

did not begin until after 2010?

**Response:** We are sorry for confusing the referee. It means that at eleven sites the measurements begin after the year 2011. We now state that "…where field sampling was carried out after the year 2011 (i.e., the years between 2012 and 2015) and/or interrupted during the period due to instrument failure (details in Table S1, Supplement)".

L271-2: There is a contradiction between a sentence that states that IASI data up until 31 December 2015 was used and the following sentence that states that data only up until 30 September 2014 was used.

**Response:** There was a wrong expression in this sentence. Actually, we used the daily IASI-NH$_3$ data from 1 January 2011 to 31 December 2015 for the spatial analysis, and from January 2011 to 30 September 2014 for temporal analysis.

We now state that "The daily IASI-NH$_3$ data (provided by the Atmospheric Spectroscopy Group at Université Libre De Bruxelles, data available at http://iasi.aeris-data.fr/NH$_3$/) from 1 January 2011 to 31 December 2015 was used for the spatial analysis in the present study. For the temporal analysis, we used the IASI_NH$_3$ from 1 January 2011 to 30 September 2014 because an update of the input meteorological data on 30 September 2014 had caused a substantial increase in the retrieved atmospheric NH$_3$ columns."

Table 1: (1) State in the caption or footnote what the significance test is testing, i.e. that it is testing for significant difference in mean concentration of a pollutant at a given site type between the northern region and the southern region. (2) The footnote should read LUY not LSY to be consistent with column heading.

**Response:** We now state in the footnote that "$^*$ and $^{**}$ denote significance at the 0.05 and 0.01 probability levels for difference in annual mean N$_r$ concentrations at a given site type between northern and southern regions, respectively."

Also, we uniformly used "LUT" as an abbreviation of land use types in the footnote and column heading.

Figure 2: The reader is referred to Table S1 in the supplement for the number of sites for each land use type in each region, but cannot the reader be directed more easily to

Table 1 in the main paper for these numbers?

**Response:** The reader cannot be directly referred to Table 1. For comparison between the periods 2011-2012 and 2013-2015, the sampling sites for land use types shown in Figure 2 have continuous 5-year (2011-2015) measurements (in total 21 sites for dry measurements, and 16 sites for wet/bulk measurements). For spatial comparisons in Table 1, the annual mean concentrations of $N_r$ species in air and precipitation for land use types were calculated based on measurements at all 27 sites.

Figure 3: (1) I assume the data shown are the means for the 5-year period, in which case it may be helpful to make this explicit in the opening sentence thus: "Seasonal mean concentrations averaged over 2011-2015 of*: : :.*". (2) As for Figure 2 (should be 3?), can the text "in Table S1 in the supplement" be replaced more directly with "in Table 1". (3) The last part of the caption should refer to significant differences between "seasons" not "sites".

**Response:** In the revised paper, we rephrased the start of caption of Figure 2 to "Seasonal mean concentrations averaged over 2011-2015 of...".

We replaced "Table S1 in the supplement" by "Table 1", as seasonal averages were calculated based on measurements at all 27 sites. Also, we changed "sites" to "seasons".

Figure 4: The same 3 comments as made above in connection with Figure 3.

**Response:** In the revised paper, we have made corresponding corrections on Figure 4 according to the referee's comments on Figure 3.

Table 2: Same comments as for Table 1.

Response: In the revision we made corresponding corrections on Table 2 according to the referee's comments on Table 1.

Figure 5: Can the reader be directed to Table 2, rather than to Table S1 in the supplement, for the number of sites of each type in each region.

**Response:** The reader cannot be referred to Table 2. For details, please see our response to similar comments on Figure 2.

Figure 7: Same comments as for Figure 3 (but with substitution of reference to Table 2 rather than to Table 1).

**Response:** In the revised paper, we made corresponding corrections on the caption of Figure 7 according to the referee's comments on Figure 3.

Figure 8: Same comments as for Figure 7.

**Response:** The reader cannot be directly referred to Table 1. Please see our response to the referee's comment on Figure 2.

L598: Rephrase start of sentence to "Eastern China is a highly industrialized: : :"

Response: Agree and done.

L 761: In comparing ion balance, presumably the (molar) concentration of NH4+ was compared against the sum of the molar concentrations of NO3- and TWICE the molar concentration of SO42-? The factor 2 is missing from the text and from the axis title of Figure 10f.

**Response:** Thank you for pointing it out. In the revised paper, we analyzed the correlation of $NH_4^+$ with the sum of $NO_3^-+2SO_4^{2-}$. Also, Figure 10f was redrawn and the corresponding sentences were changed, now read as: "At urban and rural sites, monthly mean $p$NH$_4^+$ concentrations significantly positively correlated with the sum of $p$2SO$_4^{2-}$ and $p$NO$_3^-$ concentrations (Fig. 10f). However, the slopes of regression equations between them were both smaller than unity (0.35 and 0.46 at urban and rural sites, respectively)…". In addition, we changed "Table S1" to "Table 1" in the caption of Figure 10.